# DEP-RL: Embodied Exploration for Reinforcement Learning in Overactuated and Musculoskeletal Systems

**Pierre Schumacher**[1,2]  **Daniel F.B. Haeufle**[2,3]  **Dieter Büchler**[1]  **Syn Schmitt**[3]  **Georg Martius**[1]

[1]Max Planck Institute for Intelligent Systems, Tübingen, Germany
[2]Hertie-Institute for Clinical Brain Research, Tübingen, Germany
[3]Institute for Modelling and Simulation of Biomechanical Systems, Stuttgart, Germany

## Abstract

Muscle-actuated organisms are capable of learning an unparalleled diversity of dexterous movements despite their vast amount of muscles. Reinforcement learning (RL) on large musculoskeletal models, however, has not been able to show similar performance. We conjecture that ineffective exploration in large overactuated action spaces is a key problem. This is supported by our finding that common exploration noise strategies are inadequate in synthetic examples of overactuated systems. We identify differential extrinsic plasticity (DEP), a method from the domain of self-organization, as being able to induce state-space covering exploration within seconds of interaction. By integrating DEP into RL, we achieve fast learning of reaching and locomotion in musculoskeletal systems, outperforming current approaches in all considered tasks in sample efficiency and robustness.[1]

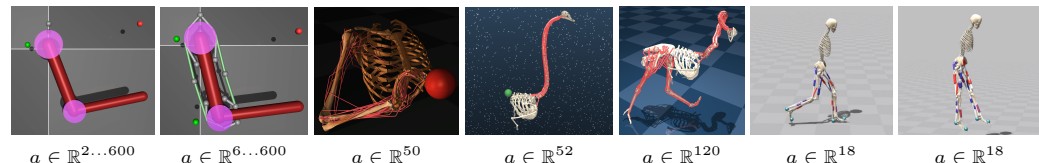

$a \in \mathbb{R}^{2\cdots600}$  $a \in \mathbb{R}^{6\cdots600}$  $a \in \mathbb{R}^{50}$  $a \in \mathbb{R}^{52}$  $a \in \mathbb{R}^{120}$  $a \in \mathbb{R}^{18}$  $a \in \mathbb{R}^{18}$

**Figure 1**: **We achieve robust control on a series of overactuated environments**. Left to right: torquearm, arm26, humanreacher, ostrich-foraging, ostrich-run, human-run, human-hop

## 1 Introduction

It is remarkable how biological organisms effectively learn to achieve robust and adaptive behavior despite their largely overactuated setting—with many more muscles than degrees of freedom. As Reinforcement Learning (RL) is arguably a biological strategy (Niv, 2009), it could be a valuable tool to understand **how** such behavior can be achieved, however, the performance of current RL algorithms has been severely lacking so far (Song et al., 2021).

One pertinent issue since the conception of RL is how to efficiently explore the state space (Sutton & Barto, 2018). Techniques like $\epsilon$-greedy or zero-mean uncorrelated Gaussian noise have dominated most applications due to their simplicity and effectiveness. While some work has focused on exploration based on **temporally** correlated noise (Uhlenbeck & Ornstein, 1930; Pinneri et al., 2020), learning tasks from scratch which require correlation **across** actions have seen much less attention. We therefore investigate different exploration noise paradigms on systems with largely overactuated action spaces.

The problem we aim to solve is the generation of motion through numerous redundant muscles. The natural antagonistic actuator arrangement requires a correlated stimulation of agonist and antagonist muscles to avoid canceling of forces and to enable substantial motion. Additionally, torques generated by short muscle twitches are often not sufficient to induce adequate motions on the joint level due to

---

[1]See https://sites.google.com/view/dep-rl for videos and code.

chemical low-pass filter properties (Rockenfeller et al., 2015). Lastly, the sheer number of muscles in complex architectures (humans have more than 600 skeletal muscles) constitutes a combinatorial explosion unseen in most RL tasks. Altogether, these properties render sparse reward tasks extremely difficult and create local optima in weakly constrained tasks with dense rewards (Song et al., 2021).

Consequently, many applications of RL to musculoskeletal systems have only been tractable under substantial simplifications. Most studies investigate low-dimensional systems (Tahami et al., 2014; Crowder et al., 2021) or simplify the control problem by only considering a few muscles (Joos et al., 2020; Fischer et al., 2021). Others, first extract muscle synergies (Diamond & Holland, 2014), a concept closely related to motion primitives, or learn a torque-stimulation mapping (Luo et al., 2021) before deploying RL methods. In contrast to those works, we propose a novel method to learn control of high-dimensional and largely overactuated systems on the muscle stimulation level. Most importantly, we avoid simplifications that reduce the effective number of actions or facilitate the learning problem, such as shaped reward functions or learning from demonstrations.

In this setting, we study selected exploration noise techniques and identify differential extrinsic plasticity (DEP) (Der & Martius, 2015) to be capable of producing effective exploration for muscle-driven systems. While originally introduced in the domain of self-organization, we show that DEP creates strongly correlated stimulation patterns tuned to the particular embodiment of the system at hand. It is able to recruit muscle groups effecting large joint-space motions in only seconds of interaction and with minimal prior knowledge. In contrast to other approaches which employ explicit information about the particular muscle geometry at hand, e.g. knowledge about structural control layers or hand-designed correlation matrices (Driess et al., 2018; Walter et al., 2021), we only introduce prior information on which muscle length is contracted by which control signal in the form of an identity matrix. We first empirically demonstrate DEP's properties in comparison to popular exploration noise processes before we integrate it into RL methods. The resulting DEP-RL controller is able to outperform current approaches on unsolved reaching (Fischer et al., 2021) and running tasks (Barbera et al., 2021) involving up to 120 muscles.

**Contribution** (1) We show that overactuated systems require noise correlated across actions for effective exploration. (2) We identify the DEP (Der & Martius, 2015) controller, known from the field of self-organizing behavior, to generate more effective exploration than other commonly used noise processes. This holds for a synthetic overactuated system and for muscle-driven control—our application area of interest. (3) We introduce repeatedly alternating between the RL policy and DEP within an episode as an efficient learning strategy. (4) We demonstrate that DEP-RL is more robust in three locomotion tasks under out-of-distribution (OOD) perturbations.

To our knowledge, we are the first to control the 7 degrees of freedom (DoF) human arm model (Saul et al., 2015) with RL on a muscle stimulation level—that is with 50 individually controllable muscle actuators. We also achieve the highest ever measured top speed on the simulated ostrich (Barbera et al., 2021) with 120 muscles using RL without reward shaping, curriculum learning, or expert demonstrations.

## 2 RELATED WORKS

**Muscle control with RL** Many works that apply RL to muscular control tasks investigate low-dimensional setups (Tieck et al., 2018; Tahami et al., 2014; Crowder et al., 2021) or manually group muscles (Joos et al., 2020) to simplify learning. Fischer et al. (2021) use the same 7 DoF arm as we do, but simplify control by directly specifying joint torques $a \in \mathbb{R}^7$ and only add activation dynamics and motor noise. Most complex architectures are either controlled by trajectory optimization approaches (Al Borno et al., 2020) or make use of motion capture data (Lee et al., 2019). Barbera et al. (2021) also only achieved a realistic gait with the use of demonstrations from real ostriches; learning from scratch resulted in a slow policy moving in small jumps. Some studies achieved motions on real muscular robots(Driess et al., 2018; Buchler et al., 2016), but were limited to simple morphologies and small numbers of muscles.

**NeurIPS challenges** Multiple challenges on musculoskeletal control (Kidziński et al., 2018; Kidziński et al., 2019; Song et al., 2021) using OpenSim (Delp et al., 2007) have been held. The top-ten submissions from Kidziński et al. (2018) resorted to complex ensemble architectures and made use of parameter- or OU-noise. High-scoring solutions in (Kidziński et al., 2019) were commonly using explicit reward shaping, demonstrations, or complex training curricula with selected checkpoints,

all of which required extensive hand-crafting. In contrast, our RL agent uses a standard two-layer architecture, no curriculum, no reward shaping, and no demonstrations.

**Large action spaces** Some studies (Farquhar et al., 2020; Synnaeve et al., 2019) tackle large action spaces by growing them iteratively from smaller versions. This would, however, require a priori knowledge of which muscle groups correspond to each other—which DEP learns by itself. Tavakoli et al. (2021) present a hypergraph-based architecture that scales in principle to continuous action spaces. Again, as it is not clear *which* muscles should be grouped, the number of possible hypergraphs is intractable. Other works (Dulac-Arnold et al., 2016; Wang et al., 2016) deal with large virtual action spaces, but only for discrete actions. Finally, studies on action priors(Biza et al., 2021; Singh et al., 2021) learn initially from expert data to bootstrap policy learning. Our method scales to large continuous action spaces and does not require demonstrations.

**Action space reduction with muscles** Several works use architectures reducing the control dimensionality before deploying RL methods. Luo et al. (2021) learn a muscle-coordination network that maps desired joint torques to muscle excitations to act on PD controllers. Jiang et al. (2019) establish a correction network that imposes output limits on torque actuators, enabling them to reproduce muscle behavior. However, these methods require specific data collection strategies and prior knowledge on joint ranges, forces or desired behaviors, before applying RL. While they simplify learning and enable large-scale control, our approach is simpler in execution, does not require knowledge about the specific muscular architecture and works with simulators of varying detail, including *elastic* tendons. Some works use PCA or other techniques to extract synergies and learn on them (Al Borno et al., 2020; Zhao et al., 2022), but they either require human data or expert demonstrations. Our approach can readily be applied to a large range of systems with only minimal tuning.

## 3 BACKGROUND

**Reinforcement learning** We consider discounted episodic Markov Decision Processes (MDP) $\mathcal{M} = (\mathcal{S}, \mathcal{A}, r, p, p_0, \gamma)$, where $\mathcal{A}$ is the action space, $\mathcal{S}$ the state space, $r : \mathcal{S} \times \mathcal{A} \to \mathbb{R}$ is the reward function, $p(s'|s, a)$ the transition probability, $p_0(s)$ the initial state distribution and $\gamma$ is the discount factor. The objective is to learn a policy $\pi(a|s)$ that maximizes the expected discounted return $J = \mathbb{E}_\pi \sum_t \gamma^t r_t$, where $r_t$ are observed when rolling out $\pi$. For goal-reaching tasks, we define a goal space $\mathcal{G}$ with $r(s, a, g)$ being 0 when $s$ reached the goal $g$ and -1 everywhere else. The policy is then also conditioned on the goal $\pi^g(a|s, g)$ following Schaul et al. (2015).

**Muscle modeling** In all our MuJoCo experiments, we use the MuJoCo internal muscle model, which approximates muscle characteristics in a simplified way that is computationally efficient but still reproduces many significant muscle properties. One limitation consists in the non-elasticity of the tendon. This reduces computation time immensely, but also renders the task fully observable, as muscle length feedback now uniquely maps to the joint configuration. Experiments with a planar humanoid were conducted in HyFyDy (Geijtenbeek, 2019; 2021), a fast biomechanics simulator that contains elastic tendons and achieves similar accuracy to the widely used OpenSim software. See Suppl. A.4 for details on the MuJoCo muscle model.

**Limitations of uncorrelated noise in (vastly) overactuated systems** The defining property of an overactuated system is that the number of actuators exceeds the number of degrees of freedom. Let us consider a hypothetical 1 DoF system, where we replace the single actuator with maximum force $F^M$ by $n$ actuators, each contributing maximally with force $F^M/n$. The force for a particular actuator is then $F_i = (F^M/n) f_i$, with $f_i \in [-1, 1]$, while the total force is $F = \sum_i^n F_i = \sum_i^n \hat{F}_i/n$. We define $\hat{F}_i = F^M f_i$ to make the dependence on $n$ clear. What happens if we apply random noise to such a system? The variance of the resulting torque is:

$$\mathrm{Var}\left(\sum_{i=1}^n \frac{\hat{F}_i}{n}\right) = \frac{1}{n^2} \sum_{i=1}^n \mathrm{Var}\left(\hat{F}_i\right) + \frac{1}{n^2} \sum_{i \neq j} \mathrm{Cov}(\hat{F}_i, \hat{F}_j). \tag{1}$$

For i.i.d. noise with fixed variance $\mathrm{Var}(\hat{F}_i) = \sigma^2$, as typically used in RL, the first term decreases with $1/n$ and the second term is zero. Thus, for large $n$ the effective variance will approach zero. The logical solution would be to increase the variance of each actuator, but as no realistic actuator can output an arbitrarily large force, the maximum achievable variance is bounded. Clearly, a vanishing effective variance cannot result in adequate exploration. For correlated noise, the second term in

Eq. 1 has $n^2 - n$ terms and decays with $1 - 1/n$, so it can avoid vanishing and might be used to increase the effective variance.

## 4 METHODS

### 4.1 DIFFERENTIAL EXTRINSIC PLASTICITY (DEP)

Hebbian learning (HL) rules are widely adopted in neuroscience. They change the strength of a connection between neurons proportional to their mutual activity. However, when applied in a control network that connects sensors to actuators, all activity is generated by the network itself and thus the effect of the environment is largely ignored. In contrast, differential extrinsic plasticity (DEP), a learning rule capable of self-organizing complex systems and generating coherent behaviors (Der & Martius, 2015), takes the environment into the loop. Consider a controller:

$$a_t = \tanh(\kappa\, C s_t + h_t), \tag{2}$$

with the action $a_t \in \mathbb{R}^m$, the state $s_t \in \mathbb{R}^n$, a learned control matrix $C \in \mathbb{R}^{m \times n}$, a time-dependent bias $h_t \in \mathbb{R}^n$ and an amplification constant $\kappa \in \mathbb{R}$. The DEP state is only constituted of sensors related to the actuators, in contrast to the more general RL state. We only consider the initialization $C_{ij} = 0$. To be able to react to the environment, an update rule is required for the control matrix $C$. A Hebbian rule proposes $\dot{C}_{ij} \propto a_{i,t}\, s_{j,t}$, while a differential Hebbian rule might be $\dot{C}_{ij} \propto \dot{a}_{i,t}\, \dot{s}_{j,t}$. The differential rule changes $C$ according to changes in the sensors and actions, but there is no connection to the consequences of actions, as only information of the *current* time step is used.

DEP proposes several changes: (1) There is a causal connection to the *future* time step $\dot{C} \propto f(\dot{s}_t)\dot{s}_{t-\Delta t}^{\top}$, where $f(\dot{s}_t) = a_{t-\Delta t}$ is an inverse prediction model and $\Delta t$ controls the time scale of learned behaviors. (2) A normalization scheme for $C$ adjusts relative magnitudes to retain strong activity in all regions of the state space $\tilde{C}_{ij} = C_{ij}/(||C_{ij}||_i + \epsilon)$, with $\epsilon \ll 1$. (3) The time-dependent bias term $\dot{h}_t \propto -a_t$ prevents overly strong actions from keeping the system at the joint limits with $\dot{s}_t = 0$. This learning rule becomes $\tau\dot{C} = f(\dot{s}_t)\dot{s}_{t-\Delta t}^{\top} - C$: the first term drives changes while the second one is a first-order decay term. The factor $\tau$ tunes the time scale of changing the controller parameters $C$. The normalization rescales $C$ at each time step. The velocities of any variable $\dot{x}_t$ are approximated by $\dot{x}_t = x_t - x_{t-1}$.

The inverse prediction model $f$ relates observed changes in $s$ to corresponding changes in actions $a$ that could have led to those changes. In the case where each actuator directly influences one sensor through the consequences of its action, the simple identity model can be used, similar to prior work (Der & Martius, 2015; Pinneri & Martius, 2018): $f(\dot{s}_t) := \dot{s}_t$. This yields the rule:

$$\tau\dot{C} = \dot{s}_t\dot{s}_{t-\Delta t}^{\top} - C. \tag{3}$$

As a direct consequence, the update of $C$ is driven by the **velocity correlation matrix** of the current and the previous state. This means that velocity correlations between state-dimensions that happen across time, for instance, due to physical coupling, are amplified by strengthening of corresponding connections between states and actions in Eq. 2. The choice of $f$ implicitly embeds knowledge of which sensor is connected to which actuator and is assuming an approximately linear relationship. This knowledge is easily available for technical actuators. In muscle-driven systems, such a relationship holds for sensors measuring muscle length with $f(\dot{s}_t) := -\dot{s}_t$, as muscles contract with increasing excitations. If there are more sensors per actuator, $f$ can either be a rectangular matrix or a more complex model that links proprioceptive sensors to actuators.

Pinneri & Martius (2018) has shown that in low-dimensional systems DEP can converge to different stationary behaviors. Which behavior is reached is highly sensitive to small perturbations. We observe that with high-dimensional and chaotic environments, the stochasticity injected by the randomization after episode resets is sufficient to prevent a behavioral deprivation. In practice, we use DEP in combination with an RL policy, such that the interplay will force DEP out of specific limit cycles, similar to the interventions of the human operator in Martius et al. (2016). Intuitively, DEP is able to quickly coerce a system into highly correlated motions by "chaining together what changes together". More details can be found in Suppl. B.3 and in Der & Martius (2015), while Suppl. C.7 contains an application of DEP to a simple 1D-system elucidating the principal mechanism.

## 4.2 INTEGRATING DEP AS EXPLORATION IN REINFORCEMENT LEARNING (DEP-RL)

Integrating the rapidly changing policy of DEP into RL algorithms requires some considerations. DEP is an independent network and follows its own gradient-free update rule, which cannot be easily integrated into gradient-based RL training. Summing actions of DEP and the RL policy, as with conventional exploration noise, leads in our experience to chaotic behavior, hindering learning (Suppl. C.2). Therefore, we consider a different strategy: The DEP policy is taking over control completely for a certain time interval, similar to intra-episode exploration (Pislar et al., 2022); RL policy $\pi$ and exploration policy Eq. 2 are alternating in controlling the system. While we tested many different integrations (Suppl. B.3 and C.2), we here use the following components:

**Intra-episode exploration** DEP and RL alternate within each episode according to a stochastic switching procedure. After each policy sampling, DEP takes over with probability $p_{\text{switch}}$. Actions are then computed using DEP for a fixed time horizon $H_{\text{DEP}}$, before we alternate back to the policy. In this way, the policy may already approach a goal before DEP creates unsupervised exploration.

**Initial exploration** As DEP is creating exploration that excites the system into various modes, we suggest running an unsupervised pre-training phase with exclusive DEP control. The data collected during this phase is used to pre-fill the buffer for bootstrapping the learning progress in off-policy RL.

## 5 EXPERIMENTS

We first conduct synthetic experiments that investigate the exploration capabilities of colored noise (Timmer & König, 1995; Pinneri et al., 2020) (see Suppl. A.2 and A.3), Ornstein-Uhlenbeck (OU) noise (Uhlenbeck & Ornstein, 1930), and DEP by measuring state-space coverage in torque- and muscle-variants of a synthetically overactuated planar arm. OU-noise, in particular, was a common choice in previous muscular control challenges (Song et al., 2021; Kidziński et al., 2018; Kidziński et al., 2019). Afterwards, DEP-RL is applied to the same setup to assure that the previous findings are relevant for learning. Finally, we apply DEP-RL to challenging reaching and locomotion tasks, among which humanreacher and ostrich-run have not been solved with adequate performance so far. Even though DEP-RL could be combined with any off-policy RL algorithm, we choose MPO (Abdolmaleki et al., 2018) as our learning algorithm and will refer to the integration as DEP-MPO from now on. See Suppl. B.1 for details. We use 10 random seeds for each experiment if not stated otherwise.

### 5.1 ENVIRONMENTS

Ostrich-run and -foraging are taken from Barbera et al. (2021), while all of its variants and all other tasks, including specifications of state spaces and reward functions, are constructed by us from existing geometrical models. We use SCONE (Geijtenbeek, 2019; 2021) for the bipedal human tasks and MuJoCo (Ikkala & Hämäläinen, 2022; Todorov et al., 2012) for the arm-reaching tasks. See Suppl. B.2 for details.

**torquearm** A 2-DoF planar arm that moves in a 2D plane and is actuated by 2 torque generators.
**arm26** torquearm driven by 6 muscles. The agent has to reach random goals with sparse rewards.
**humanreacher** A 7-DoF arm that moves in full 3D. It is actuated by 50 muscles. The agent has to reach goals that randomly appear in front of it at "face"-height. The reward function is sparse.
**ostrich-run** A bipedal ostrich needs to run as fast as possible in a horizontal line and is only provided a weakly-constraining reward in form of the velocity of its center of mass, projected onto the x-axis. Only provided with this generic reward and without motion capture data, a learning agent is prone to local optima. The bird possesses 120 individually controllable muscles and moves in full 3D.
**ostrich-foraging** An ostrich neck and head actuated by 52 muscles need to reach randomly appearing goals with the beak. We changed the reward from the original environment to be sparse.
**ostrich-stepdown** Variant of ostrich-run which involves a single step several meters from the start. The height is adjustable and the position of the step randomly varies with $\Delta x \sim \mathcal{N}(\Delta x|0, 0.2)$. The task is successful if the ostrich manages to run for $\approx 10$ meters past the step. The reward is sparse.
**ostrich-slopetrotter** The ostrich runs across a series of half-sloped steps. Conventional steps would disadvantage gaits with small foot clearance, while the half-slope allows most gaits to move up the step without getting stuck. The task reward is the achieved distance, given at the end.
**human-run** An 18-muscle planar human runs in a straight line as fast as possible. The reward is the COM-velocity. The model is the same as in the NeurIPS challenge (Kidziński et al., 2018).

**human-hop** The reward equals 1 if the human's COM-height exceeds 1.08 m, otherwise it is 0.
**human-stepdown** The human runs across slopes before encountering a step of varying height.
**human-hopstacle** The human has to jump across two slanted planes and a large drop without falling.

We observed a critical bug in ostrich-run which we fixed for our experiments. This explains the differing results for the baseline from Barbera et al. (2021) in Sec. 5.4. See Suppl. B.2 for details.

## 5.2 EXPLORATION WITH OVERACTUATED SYSTEMS

Primarily, we are interested in efficient control of muscle-driven systems. These systems exhibit a large degree of action redundancy—in addition to a myriad of nonlinear characteristics. Thus, we create an artificial setup allowing us to study exploration of overactuated systems in isolation.

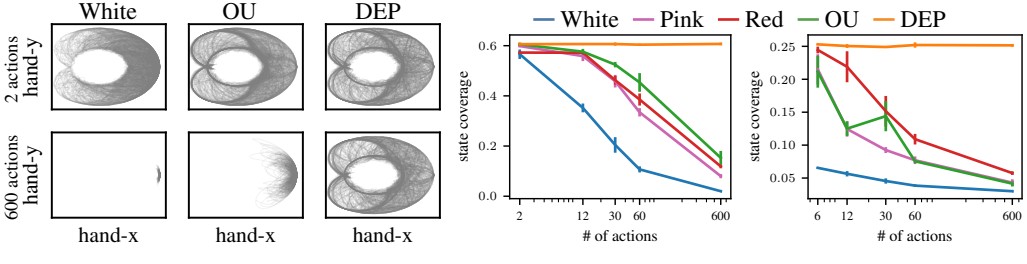

(a) torquearm trajectories    (b) torquearm state coverage    (c) arm26 state coverage

**Figure 2**: **Only DEP reaches adequate state-space coverage for all considered action spaces.** (a) Hand trajectories are collected during 50 episodes of 1000 iterations ($\Delta t = 10$ ms) of pure exploration with different noise strategies. We show hand trajectories for the original action space $a \in \mathbb{R}^2$ (top) and the expanded action space $a \in \mathbb{R}^{600}$ (bottom). (b) Endeffector-space coverage for torquearm. (c) Endeffector-space coverage for arm26.

**Overactuated torque-driven system** In the default case, the robot (Fig. 1: torquearm) can be controlled by specifying desired joint torques $a \in \mathbb{R}^2$. We now artificially introduce an action space multiplier $n$, such that the control input grows to $2n$. To apply this new control vector to the original system, we average the actions into the original dimensionality, keeping maximal torques identical:

$$a_k = \frac{1}{n} \sum_{j=1}^{n} \hat{a}_{j+n(k-1)}, \tag{4}$$

where $\hat{a}_j$ is the inflated action and $k \in [1, 2]$ for our system. We emphasize that we not only have more actions but also capture important characteristics of an overactuated system, the redundancy. As predicted in Sec. 3, we observe that redundant actuators decrease the effective exploration when using uncorrelated Gaussian (white) noise (compare 2 vs. 600 actions in Fig. 2 (a, b)). Also, for correlated pink and red colored noise or OU-noise, the exploration decays with an increasing number of actions. Only DEP covers the full endeffector space for all setups. See Suppl. A.1 for details on the used coverage metric and Suppl. C.1 for more visualizations.

**Overactuated muscle-driven system** Consider now a system with individually controlled muscle actuators (Fig. 1: arm26). This architecture is already overactuated as multiple muscles are connected to each joint, the two biarticular muscles (green) even span two joints at the same time. In addition, we apply Eq. 4 to create virtual redundant action spaces. In Figure 2 (c), we see that most noise processes perform even worse than in the previous example, even though the number of actions is identical. Only DEP again reaches full endeffector-space coverage for any investigated number of actions. These results suggest that the *heterogeneous* redundant structure and activation dynamics of muscle-actuators require exploration strategies correlated across time and across actions to induce meaningful endeffector-space coverage, which DEP can generate.

We emphasize that all noise strategies for experiments in Sec. 5.2 were individually optimized to produce maximum sample-entropy for **each** system and for **each** action space, while DEP was tuned only **once** to maximize the joint-space entropy of the humanreacher environment. We additionally observe that all strategies consequently produce outputs close to the boundaries of the action space, known as bang-bang control, maximizing the variance of the resulting joint torque (Sec. 3).

**RL with muscles**   While the previous results demonstrate that the exploration issue is significant, it is imperative to demonstrate the effect in a learning scenario. We therefore train an RL agent for differently sized action spaces and compare its performance to our DEP-MPO. Figure 3 shows that the performance of the MPO agent strongly decreases with the number of available actions, while the DEP-MPO agent performs well even with 600 available actions, which is the approximate number of muscles in the human body. DEP-MPO strongly benefits from the improved exploration of DEP. We repeated the experiment with sparse rewards and HER in Suppl. C.6, the results are almost identical.

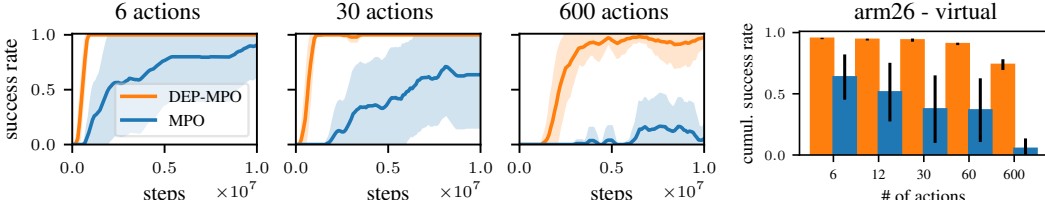

**Figure 3**: **DEP-MPO outperforms MPO in sparse point-reaching for arm26 with all virtual action spaces.** Left: Learning performance decays with a growing number of actions for MPO, DEP-MPO is largely unaffected. Right: Training-averaged success rates for different action spaces.

## 5.3   Sparse reward tasks with up to 52 actuators

We now apply our algorithm to realistic control scenarios with large numbers of muscles. As many sparse reward goal-reaching tasks are considered in this section, we choose Hindsight Experience Replay (HER) (Andrychowicz et al., 2017; Crowder et al., 2021) as a natural baseline. Generally, DEP-MPO performs much better than vanilla MPO (Fig. 4). For the more challenging environments, the combination of DEP with HER yields the best results.

DEP-MPO also solves the human-hop task, which is not goal-conditioned. As the reward is only given for exceeding a threshold COM-height, MPO **never** encounters a non-zero reward.

To the best of our knowledge, we are the first to control the 7-DoF human arm with RL on a muscle stimulation level—that is with 50 individually controllable muscle actuators. In contrast, (Fischer et al., 2021) only added activation dynamics and motor noise to 7 torque-controlled joints $a \in \mathbb{R}^7$.

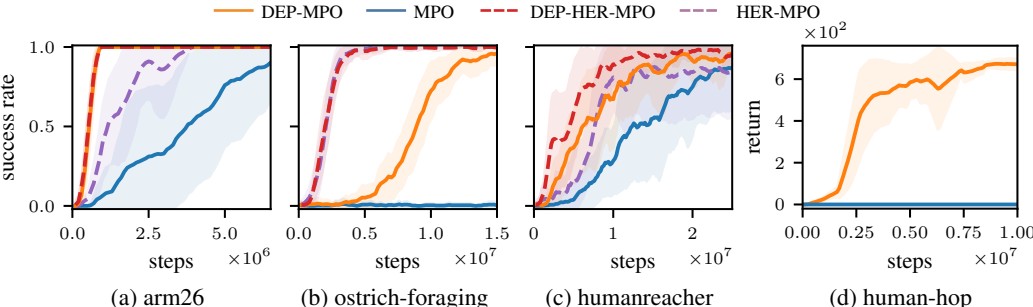

(a) arm26    (b) ostrich-foraging    (c) humanreacher    (d) human-hop

**Figure 4**: **Training performance for all sparse tasks.** (a) DEP-MPO and DEP-HER-MPO quickly solve the task. (b) While DEP-HER-MPO performs on par with HER-MPO, DEP-MPO allows the agent to solve the task in contrast to MPO. (c) DEP-MPO achieves better data-efficiency over MPO, while the best agent is DEP-HER-MPO. We conjecture that HER enables more effective use of the unsupervised data. (d) DEP-MPO finds sparse rewards even in the absence of goal-conditioning, which makes the application of HER infeasible. MPO does not encounter any non-zero reward.

## 5.4   Application to bipedal locomotion

While the results above show that DEP-RL performs well on several reaching tasks where a complete state-space coverage is desirable, it is unclear how it handles locomotion. Useful running gaits only occupy a minuscule volume in the behavioral space and unstructured exploration leads to the agent falling down very often. To test this, we apply DEP-RL to challenging locomotion tasks involving an 18-muscle human and a 120-muscle bipedal ostrich. As Barbera et al. (2021) also attempted to train

a running gait from scratch, we choose their implementation of TD4 as a baseline for ostrich-run, while we provide MPO and DEP-MPO agents. See Suppl. C.4 for additional baselines.

**High velocity running** When applying DEP-MPO to the human-run task (Fig. 5, left), we observe similar initial learning between MPO and DEP-MPO. At $\approx 10^6$ steps, DEP-MPO policies suddenly increase in performance after a plateau. This coincides with the switch to an alternating gait, which is faster than an asymmetric gait. At the end of training, 5 out of 5 random seeds achieve an alternating gait with symmetric leg extensions, while MPO only achieves this for 1 out of 5 seeds (Fig. 5, right).

In ostrich-run, we observe faster initial learning for DEP-MPO (Fig. 6, left). There is, however, a drop in performance after which the results equalize with MPO. Looking at the best **rollout** of the 10 evaluation episodes for each point in the curve (Fig. 6), we observe that some DEP-MPO trials perform much better than the average. If we consequently record the velocity of the fastest policy checkpoint from all runs for each method, we observe that DEP-MPO achieves the largest **ever** measured top speed (Fig. 7, right). The DEP-MPO policy is also characterized by overall stronger leg extension and symmetric alternation of feet positions. See Suppl. C.5 for visualizations.

In contrast, *extensive* hyperparameter tuning did not lead to better asymptotic performance for MPO (Suppl. B.6), nor did other exploration noise types (Suppl. C.4).

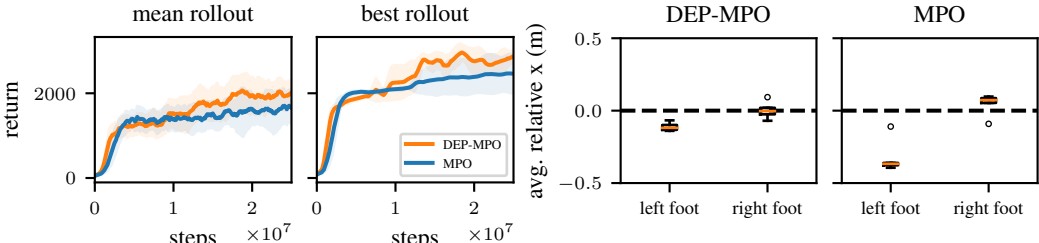

**Figure 5**: **DEP improves performance in the presence of local optima, as seen in human-run.** Left: DEP-MPO initially performs identically to MPO, before a sudden increase in performance can be observed for all trials. Right: The final gaits learned by DEP-MPO possess high symmetry, as measured by the averaged relative pelvis-deviation of the feet. Only 1 out of 5 random seeds for MPO achieves an alternating gait, which coincides with larger velocities and task returns.

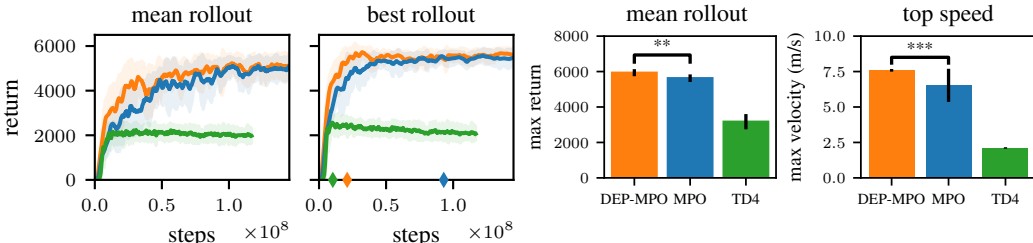

**Figure 6**: **Learning curves and maximal returns for ostrich-run.** Left: TD4 learns fast at first, but only achieves a suboptimal "hopping" gait. DEP-MPO outperforms the final MPO performance initially, but decays during late training. Even though the returns seem close, DEP-MPO achieves the fastest **ever** recorded top speed on the ostrich. Top speeds are averaged over 50 test episodes, using the fastest checkpoint for each method, marked by diamond markers. Statistical significance is marked with (**) for $p < 0.01$ and (***) for $p < 0.001$ using a student t-test. TD4 return is significantly lower than both other strategies (***, not shown for clarity of the figure).

**Robustness against perturbations** To investigate the obtained policies, we evaluate their robustness to out-of-distribution (OOD) variations in the environment. The policies are trained on a flat ground (Sec. 5.4) and then frozen, which we call in-distribution (ID). Afterwards, various OOD perturbations in the form of a large step of varying heights, a series of sloped steps or two inclined planes are introduced and the performance is measured to probe the robustness of the learned policies.

DEP-MPO yields the most robust controller against stepdown and sloped-step perturbations for all considered tasks (Fig. 7). As MPO is unable to achieve a good behavior without DEP for human-

**Table 1**: Training averaged performance metrics for the considered tasks. (S) marks success rates for reaching tasks, while (R) marks returns for all other tasks. Tasks for which HER is not applicable are marked with n.a. We perform student-t tests between the best DEP-augmented policy and the best non-DEP policy with significance levels: (*): $p < 0.05$; (**): $p < 0.01$; (***): $p < 0.001$

|  | DEP-MPO | DEP-HER-MPO | MPO | HER-MPO |
|---|---|---|---|---|
| arm26 (S) | **0.95 ± 0.01** (***) | 0.93 ± 0.01 | 0.62 ± 0.23 | 0.85 ± 0.09 |
| ostrich-foraging (S) | 0.42 ± 0.09 | **0.90 ± 0.03** (***) | 0.00 ± 0.01 | 0.88 ± 0.03 |
| humanreacher (S) | 0.72 ± 0.16 | **0.82 ± 0.12** (*) | 0.50 ± 0.23 | 0.65 ± 0.24 |
| human-hop (R) | **472 ± 85** (***) | n.a. | 0 ± 0 | n.a. |
| ostrich-run (R) | **4432 ± 702** (**) | n.a. | 4064 ± 732 | n.a. |
| human-run (R) | 1601 ± 320 | n.a. | 1395 ± 325 | n.a. |

hopstacle, we compare the performance of DEP-MPO with (OOD) and without (ID) perturbations. Interestingly, DEP-MPO is very robust, except for one random seed. This policy learned not to hop, but to move one of its legs above its shoulders, increasing its COM-height. Although hard to achieve, this behavior is sensitive to perturbations. Final policy checkpoints are used for all experiments.

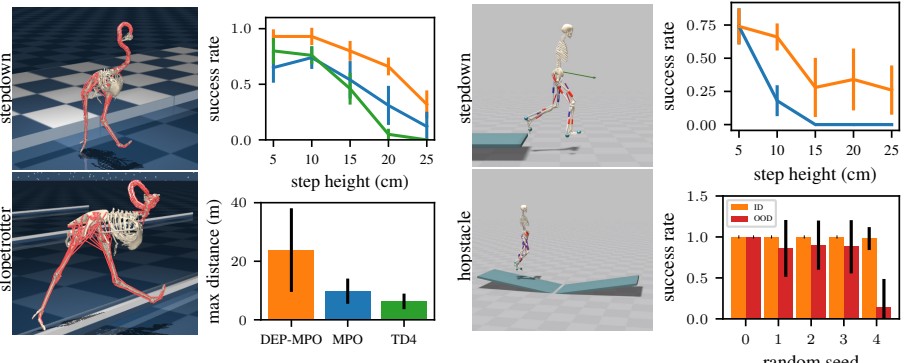

**Figure 7**: **DEP-MPO is the most robust against all considered perturbations.** Ostrich: DEP-MPO performs best under stepdown perturbations for varying step heights. The starting distance of the step was randomly varied. For the slopetrotter task, the average achieved distance is largest for DEP-MPO, although with considerable variability. Human: DEP-MPO also performs better for stepdown perturbations. For human-hopstacle, 4 out of 5 seeds of DEP-MPO achieve robust hopping. The remaining seed found a non-hopping solution achieving good returns that is not robust.

To our knowledge, we are the first to produce a robust running gait of this speed with RL applied to a system with 120 muscles, and we achieve this without reward shaping, curriculum learning or expert demonstrations. Without demonstrations, Barbera et al. (2021) only achieved a "hopping" behavior.

## 6 CONCLUSION

We have shown that common exploration noise strategies perform inadequately on overactuated systems using synthetic examples. We identified DEP, a controller from the domain of self-organization, of being able to induce state-space covering exploration in this scenario. We then proposed a way to integrate DEP into RL to apply DEP-RL to unsolved reaching (Fischer et al., 2021) and locomotion (Barbera et al., 2021) tasks. Even though we do not use motion capture data or training curricula, we were able to outperform all baselines. With this, we provide ample evidence that exploration is a key issue in the application of RL to muscular control tasks.

Despite the promising results, there are several limitations to the present work. The muscle simulation in MuJoCo is simplified compared to OpenSim and other software. While we provided results in the more biomechanically realistic simulator HyFyDy, the resulting motions are not consistent with human motor control yet. In order for the community to benefit from the present study, further work integrating natural cost terms or other incentives for natural motion is required. Additionally, the integration of DEP and RL, while performing very well in the investigated tasks, might not be feasible for every application. Thus, a more principled coupling between the DEP mechanism and an RL policy is an interesting future direction.

## 7 REPRODUCIBILITY STATEMENT

We provide extensive experimental details in Suppl. B.1, descriptions of all the used environments in Suppl. B.2 and all used hyperparameters together with optimization graphs in Suppl. B.6. The used RL algorithms are available from the TonicRL package (Pardo, 2020). The ostrich environment (Barbera, 2022) and the human-run environments are publicly available. The latter was simulated using a default model in SCONE (Geijtenbeek, 2019), an open-source biomechanics simulation software. We employed a recent version which includes a Python API, available at: `https://github.com/tgeijten/scone-core`. Additionally, we made use of the commercial SCONE plug-in HyFyDy (Geijtenbeek, 2021), which uses the same muscle and contact model as vanilla SCONE, but is significantly faster. We further include all environments and variations as well as the used learning algorithms in the submitted source code. All remaining environments are simulated in MuJoCo, which is freely available. A curated code repository will be published. We also mention hardware requirements in Suppl. B.5.

## 8 ACKNOWLEDGEMENTS

The authors thank Daniel Höglinger for help in prior work, Andrii Zadaianchuck, Arash Tavakoli and Sebastian Blaes for helpful discussions and Marin Vlastelica, Marco Bagatella and Pavel Kolev for their help reviewing the manuscript. A special thanks goes to Thomas Geijtenbeek for providing the scone Python interface. The authors thank the International Max Planck Research School for Intelligent Systems (IMPRS-IS) for supporting Pierre Schumacher. Georg Martius is a member of the Machine Learning Cluster of Excellence, EXC number 2064/1 – Project number 390727645. This work was supported by the Cyber Valley Research Fund (CyVy-RF-2020-11 to DH and GM). We acknowledge the support from the German Federal Ministry of Education and Research (BMBF) through the Tübingen AI Center (FKZ: 01IS18039B)

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

# Supplementary Material

Videos of the observed exploration patterns and learned policies are available here[2]. A curated code repository will be published upon acceptance.

## A  THEORETICAL BACKGROUND

### A.1  STATE-SPACE COVERAGE

Following (Glassman & Tedrake, 2010; Hollenstein et al., 2021), a possible measure for joint-space coverage in low-dimensional spaces can be obtained by projecting all recorded joint values $q_1^k, q_2^k, ..., q_T^k$ with $k \in \{1, 2\}$ into a discrete grid. For minimum and maximum joint values $a$ and $b$, and a grid of size $N^2$, let there be a discretized vector of values $x_k = a + k\,\Delta x$, with $\Delta x = (b-a)/N$ and $k \in \{1, ..., N-1\}$. We can then compute a matrix $S_{ij}$ such that:

$$S_{ij} = \begin{cases} 1, & \text{if } \exists t \text{ such that } x_i < q_t^0 < x_{i+1} \text{ and } x_j < q_t^1 < x_{j+1} \\ 0, & \text{otherwise} \end{cases} \quad (5)$$

The coverage is then given by:

$$\tilde{S} = \frac{\sum_{i,j} S_{ij}}{N^2}, \quad (6)$$

which is the number of visited grid points divided by the total grid size. In practice, the metric will not reach 100% as the arm cannot reach every point in space due to its geometry.

### A.2  ORNSTEIN-UHLENBECK NOISE

The OU-process (Uhlenbeck & Ornstein, 1930) is a stochastic process that produces temporally correlated signals. It is defined by:

$$x_{t+1} = x_t + \theta(\mu - x_t)\Delta t + \sigma\,\omega_t, \quad (7)$$

where $\omega_t \sim \mathcal{N}(\cdot|0, 1)$ is a noise term sampled from a standard Gaussian distribution that drives the process, $\theta$ controls the strength of the drift term, $\sigma$ the strength of the stochastic term and $\mu$ is the mean. For our experiments, we set $x_0 = \mu = 0$ s.t. $\theta$ and $\sigma$ remain as tunable parameters. In practice, actions computed by the OU-process might exceed the allowed range $a \in [-1, 1]$ for large $\sigma$; actions are subsequently clipped to the minimum and maximum values.

### A.3  COLORED NOISE

The color of random noise is defined by the frequency dependency of its power spectral density (PSD):

$$\text{PSD}(f) \propto \frac{1}{f^\beta}, \quad (8)$$

where $\beta$ is the frequency exponent of the power-law, sometimes colloquially referred to as the color of the noise. For uncorrelated, or white, noise $\beta = 0$ and the PSD is constant. In general, larger values of $\beta$ lead to noise signals with slower frequency contributions. While colors such as white ($\beta = 0$), pink ($\beta = 1$) and red ($\beta = 2$) were investigated in Sec. 5.2, we allowed any value $\beta \geq 0$ for the optimization in Sec. C.4. In practice, we use an identical implementation of colored noise to (Pinneri et al., 2020), which is based on an efficient Fourier transformation. The tuneable parameters are the color of the noise $\beta \geq 0$ and a scaling parameter $\sigma \geq 0$ which is multiplied by the noise values. Actions exceeding the allowed ranges are clipped, identical to the previous section.

### A.4  MUSCLE MODELING

The force production in biological muscles is quite complex and state-dependent (Wakeling et al., 2021; Siebert & Rode, 2014; Haeufle et al., 2014). In contrast to most robotic actuators, the force depends non-linearly on muscle length, velocity, and stimulation. The muscle also has low-pass filter characteristics, making the control problem hard for classical approaches—in addition to the typical redundancy of having more muscles than DoF. While these properties are all reproduced in the

---

[2]https://sites.google.com/view/dep-rl

MuJoCo internal muscle model, which has been used for other muscle-based studies (Barbera et al., 2021; Fischer et al., 2021; Ikkala & Hämäläinen, 2022; Richards & Eberhard, 2020), MuJoCo uses certain simplifications. It employs phenomenological force-length and force-velocity relationships which are modeled as simple functions. An additional choice, is that the tendon that connects muscles and bones is inelastic. While in more realistic systems the tendon length might vary independently of the muscle length, in MuJoCo we have:

$$l_{\text{total}} = l_{\text{muscle}} + \underbrace{l_{\text{tendon}}}_{\text{constant}}, \tag{9}$$

where $l_{\text{muscle}}$ is the length of the muscle fiber and $l_{\text{tendon}}$ the length of the tendon. Knowledge of the muscle fiber lengths $l_{\text{muscle}}$ consequently allows the unique determination of $l_{\text{total}}$ and to infer the current joint configuration. While this choice is sensible considering the immense data requirements of RL, it simplifies the control problem compared to realistic biological agents which must infer $l_{\text{total}}$ from other proprioceptive signals, which certain studies suggest to be possible in biological systems (Kistemaker et al., 2013).

We also point out that the chemical muscle activation dynamics in MuJoCo induce a low-pass filter on the applied control signals, such that temporally uncorrelated actions might not cause significant motion. The muscle activity $a_{\text{m}}(t)$ is governed by the dynamics equation:

$$\dot{a}_{\text{m}}(t) = \frac{a(t) - a_{\text{m}}(t)}{\tau(a_{\text{m}}(t), a(t))}, \tag{10}$$

where $a$ is the action as computed by the RL policy and $\tau$ is an action and activity dependent time scale, given by:

$$\tau(a_{\text{m}}(t), a(t)) = \begin{cases} \tau_{\text{act}}(0.5 + 1.5\,a_{\text{m}}(t)) & \text{if } a(t) > a_{\text{m}}(t) \\ \tau_{\text{deact}}/(0.5 + 1.5\,a_{\text{m}}(t)) & \text{if } a(t) \leq a_{\text{m}}(t) \end{cases}. \tag{11}$$

The constants $\tau_{\text{act}}$ and $\tau_{\text{deact}}$ are set to $0.01$ and $0.04$ by default, such that activity increases faster than it decreases.

See Barbera et al. (2021) and the MuJoCo documentation for more details on the muscle model and its parametrization.

Real biological systems also suffer significant delays for sensors and actuators, as well as being restricted to certain input modalities. These constraints are currently not modeled in MuJoCo.

## B EXPERIMENTAL DETAILS

### B.1 GENERAL DETAILS

For the state coverage measures (Fig. 11 and 12), we recorded 50 episodes of 1000 iterations each. The environment was reset after every episode. The state coverage metric was computed over 5 episodes at a time, after which we reset the internal state of DEP.

All experiments involving training (Fig. 3, 4 and 6) were averaged over 10 random seeds. Each point in the learning curves corresponds to 10 evaluation episodes without exploration that were recorded at regular intervals during training.

For the maximum speed measurements, the fastest checkpoint out of all runs in Fig. 6 was chosen for each method. We then executed 50 test episodes without exploration and recorded the fasted velocity within each episode.

For the robustness evaluations, the last training checkpoint of each run in Fig. 6 is chosen, as the robustness of the policies generally increases with training time in our experiments. We then record 100 episodes each with ostrich-stepdown and ostrich-slopetrotter perturbations, without any exploration. For the former, a binary success is recorded if the ostrich is able to pass the step and run for 10 additional meters afterwards. For the latter, we record the average traveled distance, as a large number of obstacles prevents most rollouts from successfully running past all of them.

For the gait visualizations (Fig. 15 and 16), the same policies as for the speed measurements are used, as they exhibit the most natural gaits. We then record a single episode and visualize the last 5 seconds to ensure a converged pattern.

## B.2 ENVIRONMENTS

All tasks except OstrichRL (Barbera et al., 2021) and the human environments Geijtenbeek (2019; 2021) were constructed from existing geometrical models in MuJoCo (Ikkala & Hämäläinen, 2022; Todorov et al., 2012) from which we created RL environments. We additionally created variants of ostrich-run involving perturbations, i.e. ostrich-stepdown, and ostrich-slopetrotter.

**torquearm**   A 2-DoF arm that moves in a 2D plane and is actuated by 2 torque generators. It is not used for RL, but as a comparative tool. Its geometry is identical to arm26, but different joint positions are reachable as it is not restricted by the geometry of the muscles. We manually restrict the joint ranges to $q_t^i \in [-120, 120]$ (degrees) to prevent self-collisions.

**arm26**   A 2-DoF planar arm driven by 6 muscles. The model was adapted from the original one in (Todorov et al., 2012), we modified the maximum muscle forces and shifted the gravity such that the arm fully extends ("down" on Fig. 1). The agent has to reach goals that are 5 cm in radius. They randomly appear in the upper right corner in a 35 cm by 15 cm rectangular area. The arm motion is restricted compared to torquearm by the passive stretch of the muscle fiber. The reward is given by:

$$r(s, s') = \begin{cases} 10, & \text{if } d(s) < 0.05 \\ -1, & \text{otherwise,} \end{cases} \tag{12}$$

where $d(s)$ is the Euclidean distance between the hand position and the goal. The episode terminates if the goal is reached, i.e. $d(s) < 0.05$. The negative reward incentivizes the agent to reach the goal as quickly as possible. Exploration in this task is difficult not only because of the overactuation, but also because the activation dynamics of the muscles require temporal correlation for effective state coverage. An episode lasts for 300 iterations, with $\Delta t = 10$ ms.

**humanreacher**   A 7-DoF arm that moves in full 3D. It is actuated by 50 muscles. The agent has to reach goals of 4 cm that randomly appear in front of it at "face"-height in a 15 cm by 30 cm by 25 cm rectangular volume. The reward function and termination condition are identical to arm26, except for the goal radius. In addition to the issues detailed in the previous paragraph, singular muscles are not strong enough to effect every joint motion. For example, pulling the arm above the shoulder requires several muscles to be stimulated at the same time, while opposing, antagonistic, muscles should not be active. The muscular geometry is also strongly asymmetric. An exploration strategy has to compute the right correlation across connected muscle groups, and across time, for each motion. The joint limits and the bone geometry create cul-de-sac states, e.g. at some point the agent might not be able to extend the elbow further to reach a goal, it has to move back and change the pose. The initial pose of the arm is fully extended and points downwards. We randomly vary the joint pose by $q_{\text{init}}^i + \mathcal{N}(0, 0.01)$ and each joint velocity by $\dot{q}_{\text{init}}^i + \mathcal{N}(0, 0.03)$ after each episode reset. This helps the RL agent and HER to make progress on the task as it causes the arm to slightly self-explore. As DEP is fully deterministic, it also prevents it from generating the same control signals during each episode in the initial unsupervised exploration phase. An episode lasts for 300 iterations, with $\Delta t = 10$ ms.

**ostrich-foraging**   This task is unchanged from (Barbera et al., 2021), except for the rewards which we modified to be sparse, identical to arm26. The termination condition is also identical. An ostrich neck and head actuated by 52 muscles need to reach randomly appearing goals with the beak. The goals appear in a uniform sphere around the beak, but only goals with goal-beak distances $d(s) \in [0.6, 0.8]$ are allowed. The goals have a radius of 5 cm. The initial pose is an upright neck position (see Fig. 1), but following the original task (Barbera et al., 2021) the pose is **not** randomized after episode resets, the last pose of the previous episode is simply kept as the first pose of the new episode. It is thus very unlikely for an agent with inadequate exploration to ever encounter a single goal. The neck itself is very flexible and offers almost no easily reachable cul-de-sac states, which we conjecture to explain the good performance of HER-MPO in Fig. 4. An episode lasts for 400 iterations, with $\Delta t = 25$ ms.

**ostrich-run**   The bipedal ostrich, from (Barbera et al., 2021), needs to run as fast as possible in a horizontal line and is only provided a weakly-constraining reward in form of its velocity. Only provided with this generic reward and without motion capture data, a learning agent is prone to local

optima. The bird possesses 120 individually controllable muscles and moves in full 3D without any external constraints. The reward is given by:

$$r(s, s') = v_x^{\text{COM}}(s), \tag{13}$$

where $v_x^{\text{COM}}(s)$ is the velocity of the center of mass projected to the x-axis. An ideal policy will consequently run in a perfectly straight line as fast as possible. The episode terminates if the head of the ostrich is below $0.9$ m, the pelvis is below $0.6$ m or the torso angle exceeds $-0.8 < \theta_{\text{torso}}(s) < 0.8$ (radians). The leg positions are slightly randomized at the end of each episode, which lasts for a maximum horizon of 1000 iterations with $\Delta t = 25$ ms.

We point out that the author's implementation of ostrich-run (Barbera, 2022) has set a default stiffness to all joints in the simulation. While this generally ensures the stability of the model, that only applies to joints that connect different parts of the system. In this case, the stiffness was also set for the root joints of the ostrich, essentially creating a spring that weakly pulls it back to the starting position. As the absolute x-position is withheld from the agent to create a periodic state input, the non-observability of the spring force destabilizes learning. We therefore explicitly set the stiffness of all root positional and rotational joints to $0$. This explains why our TD4 baseline reaches significantly higher scores than in the work by (Barbera et al., 2021). Our measured maximum return for TD4 lies at $\approx 4044$, while the reported returns without the change did not seem to exceed 2000.

**ostrich-stepdown**  A step is added to the original ostrich-run task. The height of the step is adjustable and its position randomly varies with $\Delta x \sim \mathcal{N}(\cdot|0, 0.2)$. The ostrich is initially on top of the step and has to run across the drop in height without falling over. The task is successful if the ostrich manages to run for $\approx 10$ meters past the step. The episode is terminated if the x-position exceeds $10$ m, the torso angle exceeds $-0.8 < \theta_{\text{torso}}(s) < 0.8$ rad, the head is below $0.5$ m or the head is below the pelvis height. We relaxed the termination conditions to allow for suboptimal configurations that are used to bring the ostrich back into a running pose.

**ostrich-slopetrotter**  A series of half-sloped steps is added to the original ostrich-run task. The obstacles are sloped on the incoming side while there is a perpendicular drop similar to a conventional step on the outgoing side. Rectangular stairs would disadvantage gaits with small foot clearance, while the half-slope allows most gaits to move up the step without getting stuck. There are seven obstacles spaced at $5$ m intervals in total. The episode terminates if the x-position exceeds $50$ m, the remaining termination conditions are identical to ostrich-stepdown. The obstacles are wide enough to prevent slightly diagonal running gaits from simply avoiding the obstacles. The task reward is the achieved distance, given at the end.

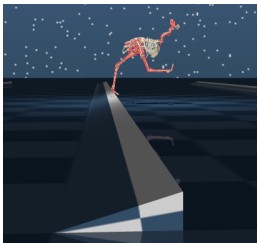

**Figure 8**: Sloped step for ostrich-slopetrotter.

**human-run**  This task uses the planar human model from the NeurIPS competition (Kidziński et al., 2018) simulated in HyFyDy instead of OpenSim. HyFyDy uses the same muscle and contact models as OpenSim, but is significantly faster. The model has 18 leg muscles and no arms. The task reward is the COM-velocity in x-direction, identical to the ostrich. The episode length is 1000. The initial position is slightly randomized from a standing position. The episode terminates if the COM-height falls below $0.5$ m.

**human-hop**  In this task, the reward of human-run is changed to be sparse. The agent receives a reward of 1 if its COM-height exceeds $1.08$ m and 0 otherwise. Periodic hopping will maximize this reward. The task is particularly challenging as there is no goal-conditioning to improve exploration in early training. The agent has to figure out a single hop from scratch. The initial state is slightly randomized from a squatting position.

**human-stepdown**  The human-run task is modified by including a parcours of varying slope with a large drop at the end. We record a success if the agent is able to navigate the entire parcours without falling to the ground.

**human-hopstacle**  The human-hop task is modified by including two inclined slopes. The task is marked as a success if the agent is able to periodically hop for 1000 time steps without falling to the

**Table 2**: Number of joints, state and action dimensions for all considered tasks. The ostrich-run variants ostrich-stepdown and ostrich-slopetrotter share identical state and action spaces, as the policies are not retrained. The planar reaching tasks torquearm and arm26 are used with virtual action spaces. For a given action multiplier $n$, we also multiply all muscle-related state data by $n$, i.e. muscle lengths, velocities, forces and activity.

|  | torquearm | arm26 | humanreacher | ostrich-foraging | ostrich-run | human-run |
|---|---|---|---|---|---|---|
| # of joints | 2 | 2 | 21 | 36 | 56 | 9 |
| action dimension | 2...600 | 6...600 | 50 | 52 | 120 | 18 |
| state dimension | 16 | 34 | 248 | 289 | 596 | 194 |
| episode length | 300 | 300 | 300 | 400 | 1000 | 1000 |
| time step | 10 ms | 10 ms | 10 ms | 25 ms | 25 ms | 10 ms |

ground. Due to the arrangement of the slopes, the agent will either hop inside the funnel, or jump out to the sides, where it will experience a steep drop.

Additional information regarding state and action sizes are summarized in Table 2. The observations are summarized in Table 3.

### B.3 DEP IMPLEMENTATION

We use a window of the recent history to adapt the DEP controller during learning. DEP requires as input 1 proprioceptive sensor per actuator. We use joint angles for the torque-driven example in Fig. 11, while all muscle-driven tasks use the sum of muscle lengths and muscle forces, normalized with recorded data to lie in $[-1, 1]$:

$$s_{\text{DEP}} = \tilde{l}_{\text{muscle}} + c\,\tilde{f}_{\text{muscle}}, \tag{14}$$

where $l_{\text{muscle}}$ is the length of the muscle fibre, $f_{\text{muscle}}$ the force acting on it and $c$ is a scaling constant. Note that $s_{\text{DEP}} \in \mathbb{R}^m$ and $a \in \mathbb{R}^n$ with $m = n$. Even though the rest of the state information is discarded, DEP computes action patterns that achieve correlated sensor changes. When alternating between DEP and the policy in DEP-RL, we also feed the current input to DEP and perform training updates. We observed performance benefits in locomotion tasks as DEP's output is strongly influenced by the recent gait dynamics induced by the policy. DEP is implemented to compute a batch of actions for a batch of parallel environments such that there is a separate history-dependent controller for each

**Table 3**: State information for all environments. The variants ostrich-stepdown and ostrich-slopetrotter use identical observations to ostrich-run. This allows the evaluation of the robustness of the trained policies against OOD perturbations.

| environment | observations |
|---|---|
| torquearm | joint positions, joint velocities, actuator positions, actuator velocities, actuator forces, goal position, hand position |
| arm26 | joint positions, joint velocities, muscle lengths, muscle velocities, muscle forces, muscle activity, goal position, hand position |
| arm750 | joint positions, joint velocities, muscle lengths, muscle velocities, muscle forces, muscle activity, goal position, hand position |
| ostrich-foraging | joint positions, joint velocities, muscle activity, muscle forces, muscle lengths, muscle velocities, beak position, goal position, the vector from beak position to the goal position |
| ostrich-run | head height, pelvis height, feet height, joint positions (without x), joint velocities, muscle activity, muscle forces, muscle lengths, muscle velocities, COM-x-velocity |
| human-run | joint positions (without x), joint velocities, muscle lengths, muscle velocities, muscle forces, muscle activity, y-position of all bodies, orientation of all bodies, angular velocity of all bodies, linear velocity of all bodies, COM-x-velocity, torso angle, COM-y-position |

**Table 4**: DEP hyperparameters for the learned policies. The test episode value signifies that an episode without DEP is recorded every N episodes. The value for arm-reaching was so large that it was effectively never used. The force scale value is used to scale the force input and the muscle length input.

(a) Arm-reaching settings.

|  | Parameter | Value |
|---|---|---|
| DEP | $\kappa$ | 1000 |
|  | $\tau$ | 80 |
|  | buffer size | 600 |
|  | bias rate | 0.00002 |
|  | s4avg | 6 |
|  | time dist ($\Delta t$) | 60 |
| integration | $p_{\text{switch}}$ | 0.01 |
|  | $H_{\text{DEP}}$ | 20 |
|  | test episode | n.a. |
|  | force scale | 0.0003 |

(b) Ostrich settings.

|  | Parameter | Value |
|---|---|---|
| DEP | $\kappa$ | 20 |
|  | $\tau$ | 8 |
|  | buffer size | 90 |
|  | bias rate | 0.03 |
|  | s4avg | 1 |
|  | time dist ($\Delta t$) | 5 |
| integration | $p_{\text{switch}}$ | 0.0004 |
|  | $H_{\text{DEP}}$ | 4 |
|  | test episode | 3 |
|  | force scale | 0.0003 |

(c) Human-run settings.

|  | Parameter | Value |
|---|---|---|
| DEP | $\kappa$ | 1896 |
|  | $\tau$ | 26 |
|  | buffer size | 200 |
|  | bias rate | 0.004154 |
|  | s4avg | 0 |
|  | time dist ($\Delta t$) | 4 |
| integration | $p_{\text{switch}}$ | 0.01 |
|  | $H_{\text{DEP}}$ | 10 |
|  | test episode | n.a. |
|  | force scale | 0.000054749 |

(d) Human-hop settings.

|  | Parameter | Value |
|---|---|---|
| DEP | $\kappa$ | 1288 |
|  | $\tau$ | 35 |
|  | buffer size | 200 |
|  | bias rate | 0.0926 |
|  | s4avg | 0 |
|  | time dist ($\Delta t$) | 6 |
| integration | $p_{\text{switch}}$ | 0.005 |
|  | $H_{\text{DEP}}$ | 30 |
|  | test episode | n.a. |
|  | force scale | 0.000547 |

environment. As the control matrix is very small, e.g. $C \in \mathbb{R}^{120 \times 120}$ even in ostrich-run, the most computationally intensive environment, this can be done with minimal overhead.

### B.4 RL IMPLEMENTATION

Our RL algorithms are implemented with a slightly modified version of TonicRL (Pardo, 2020).

### B.5 HARDWARE

Training of each DEP-MPO agent for ostrich-run, the most computationally intensive environment, was executed on an NVIDIA V100 GPU and 20 CPU cores. Training for $10^8$ iterations requires about 48 hours in real-time. Note that in general, we do train for 30 iterations for every 1000 environment interactions, which speeds up training with regard to the reported learning steps. See Sec. B.6 for details.

### B.6 HYPERPARAMETERS

We first detail the optimization choices made in the main part, before we give the specific hyperparameters that were chosen.

**Optimization for ostrich-run** We performed extensive hyperparameter optimization for the ostrich-run task with baseline MPO, but could not achieve a better final performance than default MPO parameters. The best performing set is identical in performance to the best run with default parameters

**Table 5**: RL parameters for MPO and TD4. The TD4 parameters are identical to (Barbera et al., 2021). Non-reported values are left to their default setting in TonicRL (Pardo, 2020).

(a) MPO settings.

| Parameter | Value |
|---|---|
| buffer size | 1e6 |
| batch size | 256 |
| steps before batches | 3e5 |
| steps between batches | 1000 |
| number of batches | 30 |
| n-step return | 3 |
| n parallel | 20 |
| n sequential | 10 |

(b) TD4 settings.

| Parameter | Value |
|---|---|
| buffer size | 1e6 |
| batch size | 100 |
| steps before batches | 5e4 |
| steps between batches | 50 |
| number of batches | 50 |
| n-step return | 1 |
| learning rate | 1e-4 |
| TD3 action noise scale | 0.25 |
| n parallel | 15 |
| n sequential | 8 |
| exploration | OU |
| Action noise scale | 0.25 |
| Warm up random steps | 1e4 |

**Table 6**: Baseline parameters. For HER, 80% of the time a relabelled transition is added in addition to the original one.

(a) OU-noise settings.

| | Parameter | Value |
|---|---|---|
| humanreacher | $\theta$-drift | 0.004 |
| | $\sigma$-scale | 0.02 |
| ostrich-run | $\theta$-drift | 0.1 |
| | $\sigma$-scale | 0.07 |

(b) Colored noise settings.

| | Parameter | Value |
|---|---|---|
| humanreacher | $\beta$-color | 0.04 |
| | $\sigma$-scale | 0.1 |
| ostrich-run | $\beta$-color | 0.008 |
| | $\sigma$-scale | 0.3 |

(c) Hindsight experience replay settings.

| Parameter | Value |
|---|---|
| strategy | final |
| % hindsight | 80 |

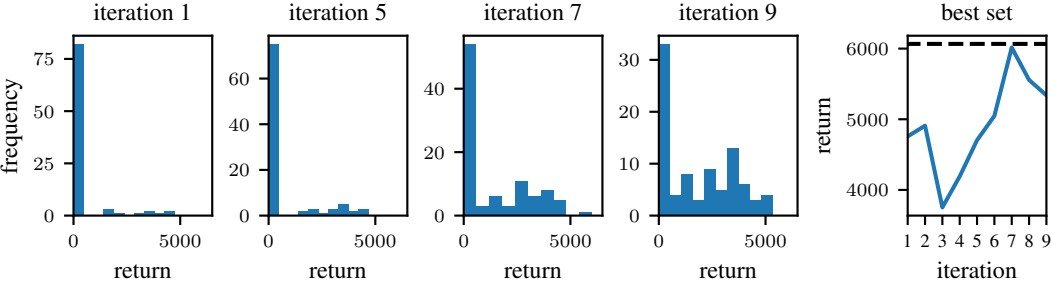

**Figure 9**: **Hyperparameter optimization for ostrich-run.** We performed 9 iterations of meta-optimization for MPO with 100 sets of parameters in each round, for a total of 900 different combinations. The final best set did not outperform the best run with the default parameters of MPO. The rightmost figure shows the best performing set for each iteration. The best return achieved over 10 evaluation episodes by MPO with default parameters out of 10 seeds is shown with a dashed black line.

in Fig. 6. In total, we computed 9 iterations of meta-optimization with 100 different sets of parameters in each round. The evolution of the performance histograms is shown in Fig. 9.

**Exploration experiments** For the experiments in Sec. 5.2, all noise strategies, with the exception of DEP, were tuned in a grid search to maximize the end effector state space coverage for each task and for each action space separately. DEP was tuned once to maximize a sample joint-pace entropy measure of the humanreacher task; its hyperparameters were then kept constant for **all** arm-reaching tasks in **all** sections of our study.

**DEP-RL** We identify three groups of tunable parameters for DEP-RL: the RL agent parameters (Table 5), the DEP parameters (Table 4), and the parameters controlling the integration of DEP and the policy. We initially optimized only for the parameters of DEP and the integration. When we afterwards ran an optimization procedure for all sets of parameters at the same time, we could not outperform our previous results. A pure MPO parameter search did also not yield better performance, such that we kept the parameters of MPO identical to the default parameters in the TonicRL library, except for minor changes regarding parallelization and batch sizes. The DEP parameters for DEP-RL in the reaching tasks were kept identical to the previous paragraph, while we heuristically chose the integration parameters. We, therefore, had 1 set of values for all arm-reaching tasks in the entire study. The DEP and the integration parameters for ostrich-run were optimized for performance, we kept them identical for ostrich-foraging.

**Baselines** The additional baselines for humanreacher and ostrich-run, see Suppl. C.4, were tuned individually for each task to maximize performance. TD4 is used with identical parameters to (Barbera et al., 2021). The values are detailed in Table 6.

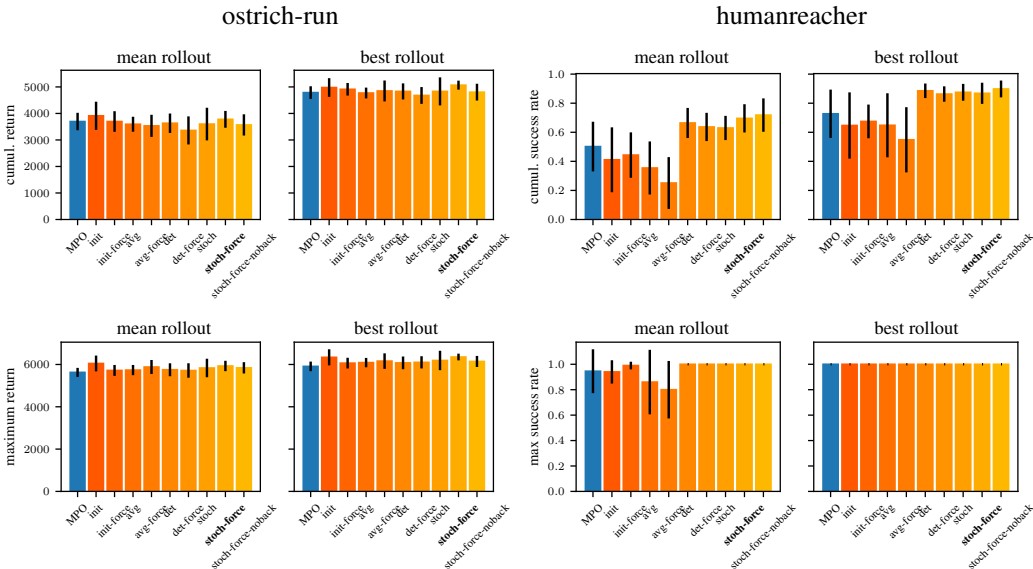

**Figure 10**: **Ablation experiments for DEP-RL.** The stoch-force-variant (bold) was used for all experiments in the main part. All ablations were trained for $5 \times 10^7$ iterations and averaged over 10 random seeds.

## C    ADDITIONAL EXPERIMENTS

### C.1    STATE-COVERAGE

We show additional visualizations of the trajectories generated by different noise processes on torquearm and arm26 in Fig. 11 and Fig. 12 respectively.

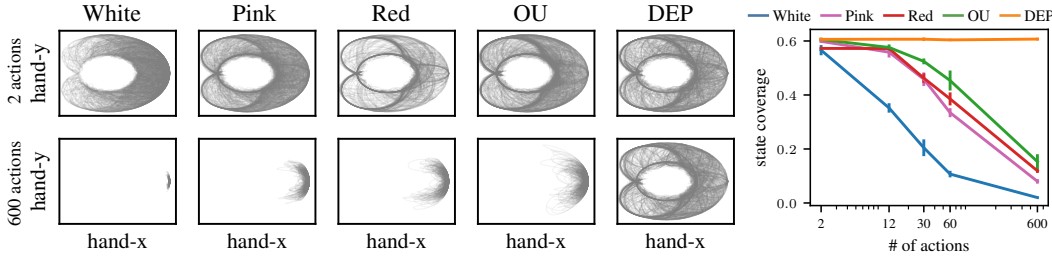

**Figure 11**: **Only DEP reaches adequate state-space coverage for all considered action spaces in torquearm.** Hand trajectories collected during 50 episodes of 1000 iterations ($\Delta t = 10$ ms) of pure exploration with different noise strategies. Left: Hand trajectories for the original action space $a \in \mathbb{R}^2$ (top) and expanded action space $a \in \mathbb{R}^{600}$ (bottom). Right: Endeffector-space coverage.

### C.2    ABLATIONS

We experiment with several implementations of DEP-RL and show cumulative and maximum performances for a selection of them in Fig. 10. The ablations are:

**init**  DEP is only used for initial unsupervised exploration. The collected data is used to pre-fill the replay buffer. This component is active in all other ablations.

**avg**  DEP actions and policy actions are combined in a weighted average. The DEP weight is much smaller than the policy weight.

**det**  DEP and the policy control the system in alternation. They are deterministically switched s.t. DEP acts for $H_{\mathrm{DEP}}$ iterations and the RL policy for $H_{\mathrm{RL}}$ iterations. The current state is used to train and adapt the DEP agent, even if the RL action is used for the environment. DEP actions are also added to the replay buffer of the RL agent.

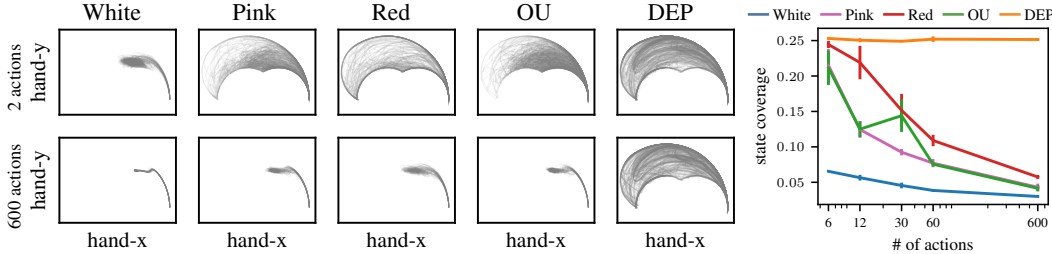

**Figure 12**: **Only DEP reaches adequate state coverage for all considered action spaces in arm26.** Hand trajectories collected during 50 episodes of 1000 iterations ($\Delta t = 10$ ms) of pure exploration with different noise strategies. Left: Hand trajectories for the original action space $a \in \mathbb{R}^6$ (top) and expanded action space $a \in \mathbb{R}^{600}$ (bottom). Right: endeffector-space coverage.

**stoch** Identical to the previous ablation, but the alternation is stochastic s.t. there is a probablitiy $p_{\text{switch}}$ that DEP takes over for $H_{\text{DEP}}$ iterations.

We additionally introduce force-variants of all these ablations where the state input of DEP is not only composed of the muscle lengths, but also the forces acting on the muscles. We observed that this causes DEP to seek out states that produce more force variations, which are generally interesting for locomotion. Lengths and forces are normalized from recorded data and then added together with a certain weighting, in order to not change the number of input dimensions of DEP.

Lastly, we show the performance for a *noback*-variant, where DEP is not learning in the background while the RL agent is taking over control. Even though the init-variant achieves a fast gait for locomotion, we chose the **stoch-force**-variant for all the results in the main section, as it achieves good performance on all tasks. All ablations were averaged over ten random seeds.

### C.3 ACTION CORRELATION MATRIX

We recorded action patterns generated from different noise strategies applied to ostrich-run. Even though we only recorded 50 s of data, and DEP was learning from **scratch**, strong correlations and anti-correlations across muscle groups can be observed in Fig. 13. We deactivate episode terminations in order to observe the full bandwidth of motion generated by DEP. For this particular task, the ostrich was lying on the ground while moving the legs back and forth in an alternating pattern. Uniform, colored and OU noise are unable to produce significant correlations across actions.

### C.4 ADDITIONAL BASELINES

We combine MPO with colored ($\beta$-MPO) and OU-noise (OU-MPO) by summing them to the action computed by the baseline MPO policy. We then apply these new algorithms to humanreacher and ostrich-run, as they constitute challenging reaching and locomotion tasks. We also tested an implementation of DEP-TD4 on ostrich-run. The base agents were identical for all baselines, while the OU and the colored noise were optimized to achieve the best performance, see Suppl. B.6. It can be seen in Fig. 14 (left) that DEP-MPO achieves the largest returns in ostrich-run, while OU-MPO intermittently outperforms vanilla MPO. Similarly, OU-MPO and $\beta$-MPO perform better than MPO in the humanreacher task, as seen in Fig. 14 (right), but DEP-MPO achieves the best performance.

### C.5 OSTRICH GAIT VISUALIZATION

We show the achieved foot movements and footstep patterns of the ostrich for the different algorithms in Fig. 15. The leg deviation is strongest for DEP-MPO, while it also achieves the most regular foot pattern. This suggests that DEP improves exploration, as it allows for policies that utilize the embodiment of the agent to a greater extent, while also achieving larger running velocities. MPO manages less leg extension, while the TD4 gait is irregular and asymmetric. The step lengths of $\approx 1$m achieved by DEP-MPO are also quite close to real ostriches (Rubenson et al., 2004), while MPO and TD4 only achieve small step lengths of $\approx 0.5$m and $\approx 0.3$m respectively. We provide additional visualizations of the relative $x$ and $z$ trajectories of the feet during locomotion for each algorithm in Fig. 16.

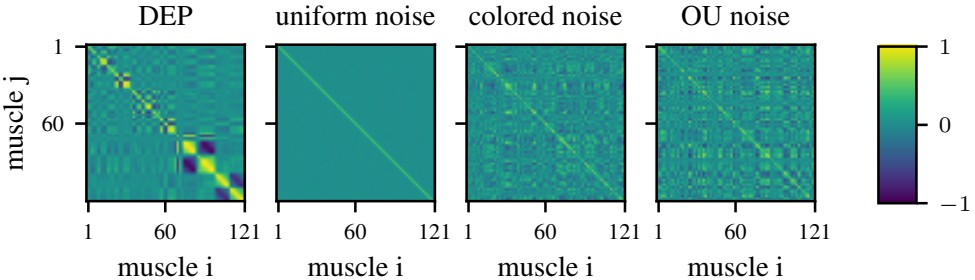

**Figure 13**: **Action correlation matrix for different exploration strategies.** We recorded 50 s of data (1000 transitions) from ostrich-run and computed the correlation matrix from the action trajectories. Note that even though DEP was initialized with $C_{ij} = 0$, strong correlation and anti-correlation patterns can be observed for antagonistically opposed muscle groups. Colored and OU noise do not exhibit strong correlations across actions, as they are designed to produce temporally correlated signals.

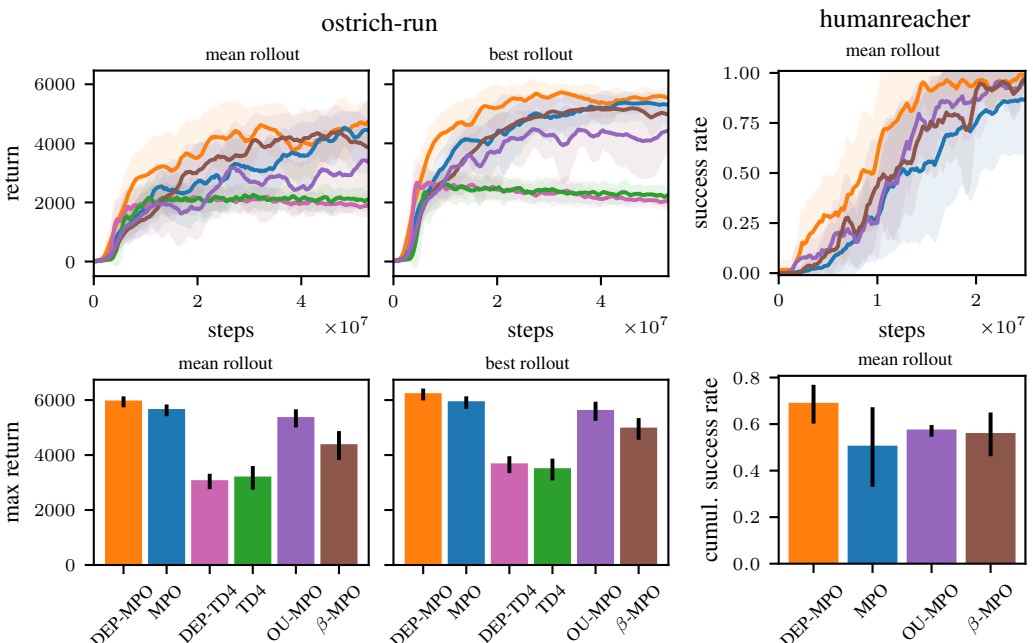

**Figure 14**: Left: Additional baselines for ostrich-run. We provide OU and $\beta$-MPO agents by summing them to the action computed by MPO as with regular exploration noise. Right: Identical baselines for humanreacher. OU and colored noise processes were optimized for the present tasks, while the base MPO agent was identical for **all** experiments in this figure.

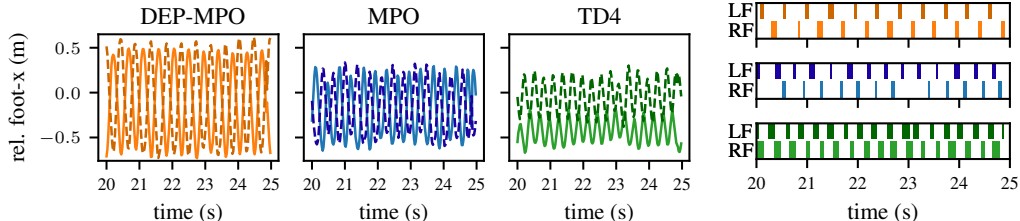

**Figure 15**: **DEP-MPO achieves the most widespread and symmetric gait.** Left: The relative sagital foot position w.r.t. the torso visualizes the leg extension during locomotion. DEP-MPO creates a symmetric gait with $\approx$ 1m step length. For MPO the step length is much shorter. TD4 has a completely shifted gait, the left foot is often in front of the right foot. Right: Foot contact pattern for all gaits. The shaded areas mark the time during which the respective foot (LF: left foot or RF: right foot) is in contact with the ground. Visualized are the last 5 seconds of an evaluation episode to ensure a converged pattern.

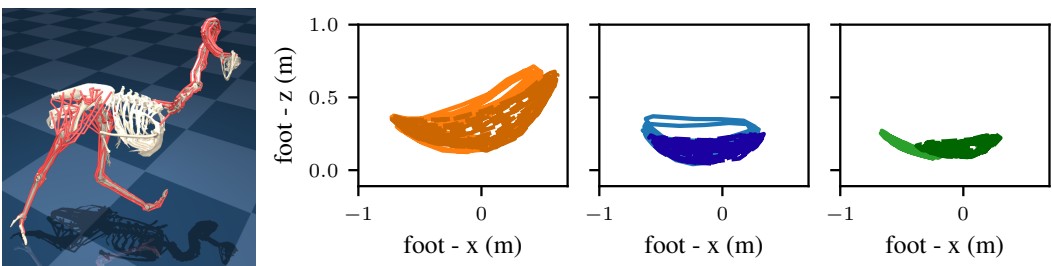

**Figure 16**: Foot gait patterns for ostrich-run during the final 5 s of an episode. While DEP-MPO portrays a slight asymmetry in the z-direction, MPO and TD4 are noticeably less symmetric.

### C.6 Additional evaluation with dense and sparse rewards in large virtual action spaces

We present in Fig. 17 the results for experiments with virtual action spaces, as shown in Fig. 3, but here in addition with HER and for the case of sparse and dense rewards. For dense rewards (top), an increasing number of actions requires more environment steps for vanilla MPO to solve the task, while the performance collapses for 600 actions. DEP-MPO reaches good performance for each considered action space. MPO performs similarly in the sparse task, albeit a larger variation across runs can be observed. While HER elicits faster learning, it still requires significantly more environment steps for 6 and 120 actions than DEP-MPO and does not achieve good performance for 600 actions. DEP-MPO and HER-DEP-MPO quickly solve all tasks.

### C.7 Description of a one-dimensional system

In this section, we give an outline of how DEP works in a theoretical scenario. We will first describe a simplified DEP rule and detail how its dynamics might excite the mountain car system (Moore, 1990) to cover the state. We will then apply the simplified rule on the mountain car environment and present the results, see Fig. 19.

**Mountain car** The original DEP controller is described by:

$$a_t = \tanh(C s_t + h_t), \tag{15}$$

with the state $s_t \in \mathbb{R}^n$, a time-dependent bias $h_t \in \mathbb{R}^m$, the action $a_t \in \mathbb{R}^m$ and the learned control matrix $C \in \mathbb{R}^{m \times n}$. The update rule is now defined as:

$$\tau \dot{C} = f(\dot{s}_t) s_{t-1}^\top - C, \tag{16}$$

where $f(\cdot) : \mathbb{R}^m \to \mathbb{R}^n$ is an inverse model, relating future changes in the state to the change in action that caused them. First, we state the assumptions for this section:

1. We do not consider the normalization scheme for $C$.

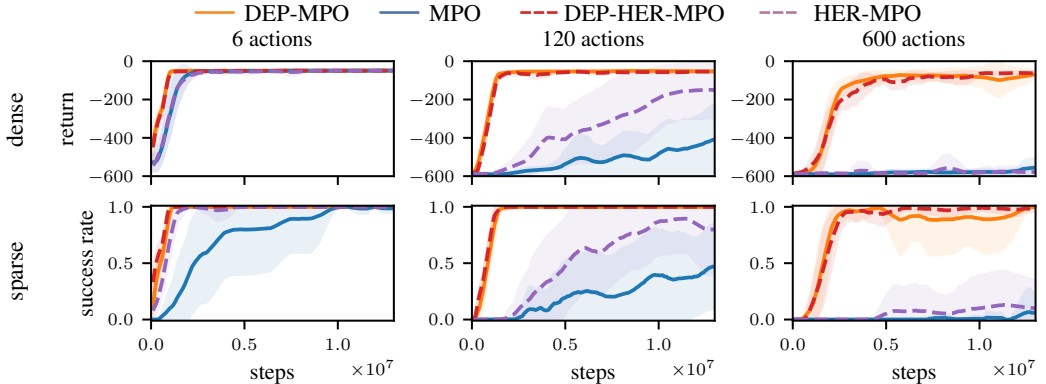

**Figure 17**: **DEP-MPO outperforms MPO in sparse and dense reward point-reaching for arm26 with all virtual action spaces.** Top: Learning performance for dense rewards. DEP-MPO strongly outperforms MPO, no significant movement learning could be detected for MPO with 600 actions. Bottom: Success rates for sparse reward reaching. While HER seems to increase the performance of the MPO baseline in most cases, the success rate only increases marginally for 600 actions, even after almost $1.5 \times 10^7$ steps. DEP-MPO solves all tasks with or without the addition of HER.

2. We choose $f(\dot{s}_t) = \dot{s}_t$

3. A bias is not considered $h_t = 0$

4. The nonlinearity is approximated by the first term of a Taylor expansion $\tanh(x) \approx x$.

5. We consider $C$ to instantaneously fulfill the update rule, without considering update dynamics.

Combining all the assumptions, we obtain:

$$C = \dot{s}_t \dot{s}_{t-1}^{\top}. \tag{17}$$

We will now consider a simple system with 1 sensor: the continuous action mountain car.

The original environment defines the RL state $s_t^{\mathrm{RL}} = (x_t, \dot{x}_t)$. From this we extract the sensor information for DEP $s_t = x_t$, with $s \in \mathbb{R}$. In this 1-sensor formulation, the velocity correlation matrix in Eq. 17 is a scalar, as there is only 1 sensor. Consequently, also the $C$ matrix is scalar. The resulting control equation is:

$$a_t = C s_t = \dot{s}_t \dot{s}_{t-1} s_t. \tag{18}$$

Let us assume an initial state of the car slightly to the right side of the valley, as in Fig. 18, with a positive initial velocity. States $s > 0$ are positions to the right of the valley bottom, while $s < 0$ to the left. The initial velocity will cause the car to move up the mountain. After some time, the velocity correlation will be $\dot{s}_t \dot{s}_{t-1} > 0$ with $s > 0$, leading to $C s_t = a_t > 0$. Logically, the car then starts pushing to the right, reinforcing the movement pattern and trying to increase its velocity. However, the task is set up such that the force is insufficient to directly go up the mountain. It will thus change the movement direction at some point and reverse due to gravity.

After changing direction, $\dot{s}_t \dot{s}_{t-1}$ will reverse sign as the *previous* velocity still points to the right, while the *current* velocity points to the left. Thus, $\dot{s}_t \dot{s}_{t-1} < 0$ with $s > 0$, and $C s_t = a_t < 0$. The car will consequently try to push into the negative direction, accelerating downwards. If the past sensor derivative $\dot{s}_{t-1}$ is defined as only one time step away, this trend will immediately reverse and the car will decelerate.

If, however, $\dot{s}_t$ is chosen to be not one time step apart from $\dot{s}_{t-1}$, but $\Delta t \in \mathbb{N}$ steps, then it will take several time steps until the sign reversal of $\dot{s}_t \dot{s}_{t-\Delta t}$ happens. In this intermittent regime, $\dot{s}_t \dot{s}_{t-\Delta t} < 0$ with $s > 0$, and $C s_t = a_t < 0$, pushing the car to the left, until the velocity correlation changes sign again.

If $\Delta t$ is appropriately chosen, however, by the time the reversal happens, the car will have moved into the negative state region $s < 0$. In this new region, $\dot{s}_t \dot{s}_{t-\Delta t} > 0$ with $s < 0$, and $C s_t = a_t < 0$, which pushes the car further to the left and up the mountain, until the phenomenon repeats.

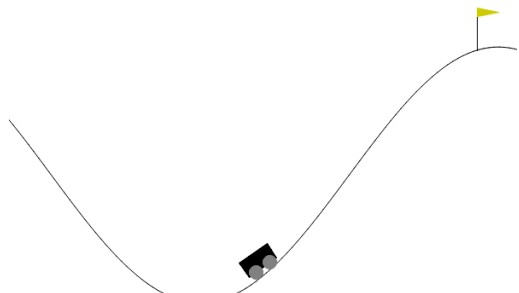

**Figure 18**: Continuous-action mountain car environment. To solve the task, the car must be brought to the top of the right hill. However, the motor is too weak for a direct approach. The solution involves pushing the car from left to right and vice-versa with the correct frequency to gather momentum and climb up the mountain.

We offer simulation results of the described dynamics in Fig. 19. For this simulated example, we consider again the nonlinearity of $\tanh$ (Eq. 15), as it allows us to multiply $C$ by a large constant and still satisfy the action limits of the environment. This is a necessary step as we omitted the normalization scheme for $C$. We thus consider $C = \tanh(\kappa \dot{s}_t \dot{s}_{t-\Delta t})$, with $\kappa \gg 1$.

To demonstrate the influence of $\Delta t$, we plot the results for two examples (Fig. 19). For $\Delta t = 27$, the system trajectory at time step $\approx 95$ shows a reversal point. Before this reversal point, the position $s$ is positive and so is the correlation $\dot{s}_t \dot{s}_{t-\Delta t}$, which leads to positive actions. Due to gravity and the weak motor, however, the car starts to move into the opposite direction. As there was a velocity sign change, we have $\dot{s}_t \dot{s}_{t-\Delta t} < 0$ while the position $s_t$ is still positive, which yields a negative action: The car is accelerating downwards and builds up momentum, which, after a few repetitions, allows it to explore the environment.

The values of $\Delta t$ for this example were chosen to yield good visualization. We observe full exploration of the mountain car system for $\Delta t \in \{5, ..., 28\}$.

This might seem like an overly simplified example, but it shows how DEP can increase the variance in a sensor value, even if a large number of actuators is associated to it. Empirical evidence additionally demonstrates that in high-dimensional systems, if all other DEP components are considered, DEP becomes less sensitive to the exact system dynamics and parameter specifications, generalizing easily to muscle-driven systems with over 120 muscles. Note that as the mountain car task contains an harmonic potential well, ubiquitous in many physical systems, the analysis might hold for a wide range of models, even considering elastic muscles with nonlinear spring elements.

## C.8 SENSITIVITY TO CHANGES IN $p_{\text{switch}}$

We performed an ablation study over the parameter $p_{\text{switch}}$, that controls the probability of switching from the RL policy to the DEP controller. Figure 20 shows that the training averaged success rates for DEP-MPO are much less sensitive to this hyperparameter for the humanreacher task than the average returns for ostrich-run. We conjecture that locomotion is inherently more unstable and that higher DEP probabilities cause the agent to fall down often, which hurts learning performance.

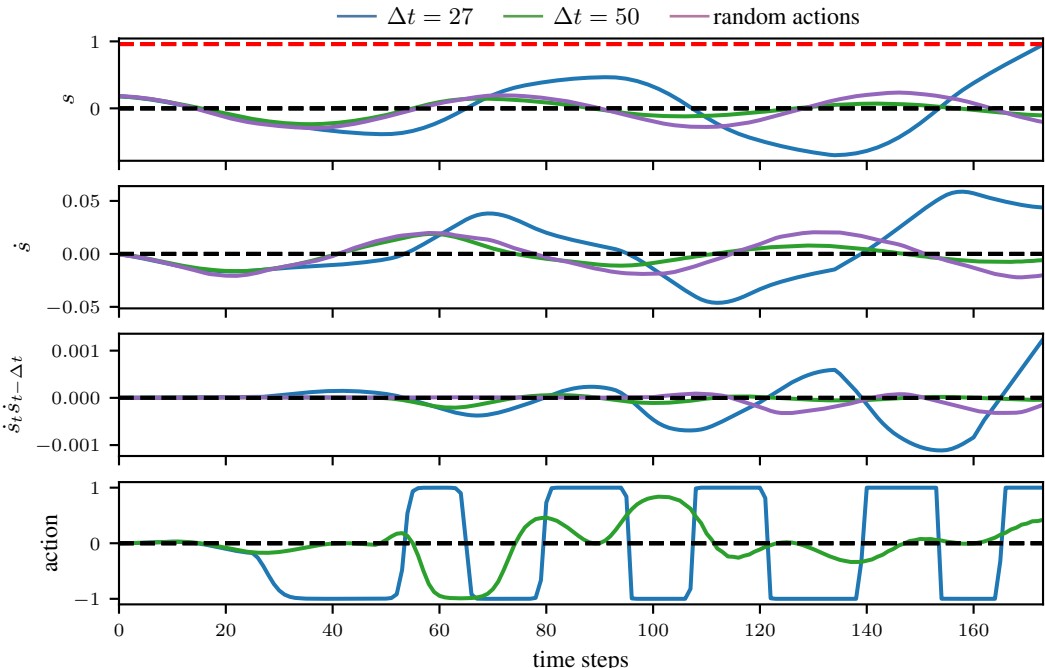

**Figure 19**: A simplified DEP rule in the mountain car environment (Moore, 1990). The red line marks the position threshold at which the task is solved. The black lines mark the zero point. Negative values of $\dot{s}_t \dot{s}_{t-\Delta t}$ mark intermittent regimes where the controller output reverses, eventually bringing the system into coherent motion. For $\Delta t = 27$, the reversal of $\dot{s}_t \dot{s}_{t-\Delta t}$ happens on a time scale that is able to excite the system, while the setting $\Delta t = 50$ is not able to induce sufficient exploration. The random actions are drawn from a standard Gaussian $\mathcal{N}(0, 1)$. We do not show the proposed actions for the Gaussian to keep the figure readable. The position $s$ for the setting $\Delta t = 27$ clearly oscillates with larger and larger amplitudes over time, increasing the effective variance of the sensor value. These two values of $\Delta t$ were chosen to yield good visualizations, we observe full exploration of the mountain car system for values $\Delta t \in \{5, ..., 28\}$.

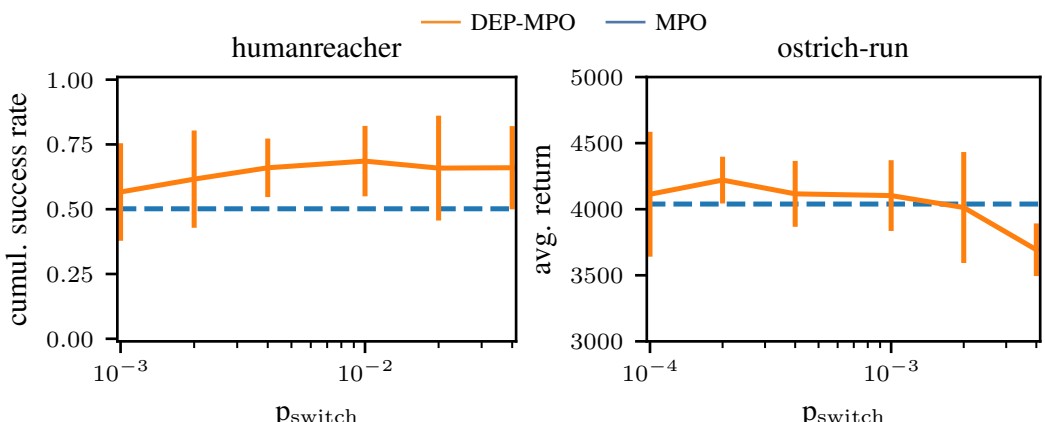

**Figure 20**: **Ablation over the DEP probability with DEP-MPO for two tasks.** Left: Humanreacher performance is overall robust for different settings of $p_{\text{switch}}$, while the benefit of DEP disappears for very small values. Right: Ostrich-run is more sensitive to the parameter, as locomotion is generally more unstable. Large DEP probabilities cause the agent to fall down very often. The horizontal line marks the average MPO performance without DEP. All values are averaged over 5 seeds. Note the log-axis.

