# OpenReview forum: "DEP-RL: Embodied Exploration for Reinforcement Learning in Overactuated and Musculoskeletal Systems"
_ICLR.cc/2023/Conference — ICLR 2023 notable top 25%_

### Official Review · Reviewer_Bv9J · 2022-10-20

**Confidence:** 4
**Correctness:** 4
**Technical Novelty And Significance:** 4
**Empirical Novelty And Significance:** 4
**Recommendation:** 10

**Clarity, Quality, Novelty And Reproducibility:**

The method is sound, novel and well-motivated.

Details are provided for reproducibility and the authors also mention the code will be released.

**Strength And Weaknesses:**

Pro:

1. A novel combination of RL and DEP for effective learning for muscle-driven systems. Experiments demonstrate the proposed approach outperforms existing baselines.

2. The method is well motivated by issues surrounding the difficulty to explore with overactuated system. I especially appreciate the simple example that is introduced to provide intuition.

Con:

Not much. To nitpick a little, the motion quality for the ostrich control is still not natural.

**Summary Of The Paper:**

This paper combines differential extrinsic plasticity (DEP) and reinforcement learning for more efficient learning of how to control overactuated systems. Demonstrations are done on simple hand design systems as well as complicated muscle-driven locomotion control.

**Summary Of The Review:**

This provides a new and intuitive explanation of why it is hard to learn controllers for overactuated systems with standard RL approaches using Gaussian noise. DEP is introduced to effectively resolve this issue. A wide range of examples is used to demonstrate the effectiveness of the proposed method,  including state coverage of a simple overactuated system and challenging locomotion control with bipedal systems with many muscles. I believe this paper makes a big step forward in solving muscle-driven control problems.

---

> ### Author Response · Authors · 2022-11-15
> **Response to Bv9J**
>
> We thank the reviewer for the extremely positive response and are happy that the simpler experiments are also appreciated!
>
> > To nitpick a little, the motion quality for the ostrich control is still not natural.
>
> We also agree with the lack of naturality of the ostrich, especially the neck. In a sense, DEP-RL does not induce natural motions per se, it merely improves exploration to show the agent which behaviors could achieve a larger return—leading in this case to a larger top speed. It seems that with the purely velocity-based reward function and the straight terrain, the precise neck pose is not as important, as long as it allows for a stable gait. So, while we demonstrate an improvement in exploration and performance for the control of these complex systems, future work should address how to achieve more natural behaviors to reproduce motion data from real animals.
>
> We thank you again for the kind words, highlighting the significance of our work for muscle-driven control.

---

### Official Review · Reviewer_sbak · 2022-10-23

**Confidence:** 4
**Correctness:** 4
**Technical Novelty And Significance:** 4
**Empirical Novelty And Significance:** 4
**Recommendation:** 8

**Clarity, Quality, Novelty And Reproducibility:**

The paper is generally very clearly written, within the available space constraints.

It builds on the previously introduced Differential Extrinsic Plasticity (DEP).  In this paper it really becomes a lot more useful via the integration with RL.  Thus I see it as being highly original.

**Strength And Weaknesses:**

Strengths:

- important and difficult problem:  high-D action spaces, redundant actuation, musculoskeletal control
- impressive results and progress on the above problem
- progress towards understanding how overactuation might be resolved in natural systems

Weaknesses:
- missing possible intuition about how the redundant actuation is ultimately resolved
- writing: The underlying assumptions needed to apply DEP could be articulated more clearly,
  when coming from the common RL mindset, where there is generally great freedom in the state specification.
- mixed motion quality of some of the results, subjectively speaking, e.g., Ostrich running, Ostrich head-and-neck during gaits
- unclear how more classical biomechanics criteria, i.e., metaboloic energy consumption, might come into play,
  or if these results are predictive in any way of what can be observed in natural systems


**Summary Of The Paper:**

Conventional exploration strategies for RL scale poorly to high-dimensional action spaces.
Differential Extrinsic Plasticity (DEP), integrated with RL, is proposed as a mechanism for coping
with highly-redundant actuation spaces, such as that of multiple musculotendon units that span joints
in biomechanical models. The key idea of DEP is to learn a state-to-action mapping that is largely defined
by a linear state-to-action matrix, C, which is updated by the velocity correlation matrix of the current
and previous state.  This induces its own dynamics, without a goal task, and has been shown in prior
work on self-organizing behavior, to produce converge to useful stationary behaviors.  In this work,
it is proposed to integrate this with RL, via alternating DEP and RL motion phases during learning.
The results are tested on a wide variety of scenarios and over-actuated systems. Various baselines
and ablations are presented, including Uhlenbeck-Ornstein noise, colored noise, MPO and TD4 results,
integration with HER, and more.


**Summary Of The Review:**

The paper makes significant progress on tackling an important and difficult problem,
namely that of high-dimensional action spaces that arise in over-actuated systems
such as musculoskeletal models. It makes a very valuable contribution in this regard.

The underlying assumptions needed to apply this framework could be articulated a bit more clearly,
when coming from the common RL mindset, where there is generally great freedom in the state specification.
The paper is in a good position to also further state caveats and differences in terms of bridging from RL to
the biomechanics community, which may have different evaluation criteria, related to motion quality,
experimental comparisons to ground-truth data, and predictive capabilities.

-------------------------------------------------------------
Detailed comments follow below.

Given the importance of matrix C, perhaps give it a name that connects with its implied role?
Conceptually, I now see that it represents the weights of a one-layer feedf-foward network.

The original DEP paper discusses the role of updating the threshold terms, "to
drive the neurons away from their saturation regions."  Is this still important in this work, to stay
away from converging to static poses, or does the integration with RL obviate this need?

Perhaps some of the intuition provided in the original DER paper could be reiterated in this paper, to help
the reader, i.e., “chaining together what changes together.”

Is the value of Delta t kept fixed for the bulk of the results?

The coverage of the related work is already excellent in the current paper, although
there is additional early work on muscle-based locomotion that also comes to mind, e.g.,
"Optimizing Locomotion Controllers Using Biologically-Based Actuators and Objectives" (2012)
"Flexible Muscle-Based Locomotion for Bipedal Creatures" (2013),
which are both objective-based optimizations, albeit with strong priors on the control architecture.

To what extent can the final structure of the matrix C be interpreted?
This would be useful to here about. Is there an emergent low-dimensional structure?
Is this structure consistent across multiple runs?

For muscles spanning multiple joints, the inverse mapping, f, presumably becomes much less clear,
as it now also depends on the motion of more than one joint.  Does this have any impact on the
ability to achieve good results?  Relatedly, I'm also curious to better understand how critical
it is to have knowledge of the 1:1 correspondence between sensors and actuators.

4.1 DEP introduction

It is not clear in the description at this point if the state s can be very general, as in RL,
or if it refers much more specifically to the length of the muscle actuators.
I believe that it is the latter, so it is worthwhile clarifying.

Text after eqn (2): "f is an inverse model"
This begs the question "an inverse model of what?".

Update rule for C dot, with "-C" being a damping term. Perhaps call this a first-order decay term,
which presumably acts as a regularizer?

What forms of perception are available for the last three ostrich tasks, i.e., foraging, stepdown, slopetrotter?
Perhaps it is in the appendix, but providing basic knowledge of "is the their knowledge fo the upcoming terrain"
would be useful.

Appendix A "... chemical muscle activiation dynamics"
It's great to see this discussed, are these actuation dynamics being modeled or not?
Presumably not (which is fine as a simplification), but state this explicitly.

DEP-MPO for the human hopping is, subjectively speaking for this reader, the highest quality result. Very nice.

---

> ### Author Response · Authors · 2022-11-15
> **Response to sbak (1/2)**
>
> We are impressed with this extremely detailed  review. It raises a few excellent points.
>
> > The underlying assumptions needed to apply this framework could be articulated a bit more clearly, when coming from the common RL mindset
>
> This is certainly true, we changed the DEP introduction to make this point a bit clearer: The RL state can be arbitrarily chosen, but the DEP state is a vector of actuator sensor values. In the robotic case, this might be a joint position for each motor that actuates a joint, while in the muscle case, this is the length of each muscle.
>
> > matrix C, perhaps give it a name that connects with its implied role?
>
> We changed the name to control matrix in the text.
>
> > The original DEP paper discusses the role of updating the threshold terms
>
> While we retain the bias-mechanism whose role it is to prevent saturation of the actions, we use relatively small values for its strength (0.0002, compared to ~0.1 in the original DEP paper). Empirically, we observe that without it, the agent might sometimes get stuck with an extended arm. However, the slight initial state randomization and the interplay with the RL agent largely mitigate this problem. See also **Fig. 19 (Supp.)**, where the mountain-car problem is completely explored with a simplified DEP-variant, which does not contain a bias-mechanism. We conjecture that the approximately quadratic potential of the mountain (present in muscles as tendon elasticity) might overcome the need for a bias term.
>
> > Perhaps some of the intuition provided in the original DER paper could be reiterated
>
> We added the relevant expression to the main  text.
>
> > additional early work on muscle-based locomotion that also comes to mind
>
> We thank you for these paper suggestions. While these are very interesting publications, they focus strongly on the naturalness of the resulting motions. We see our submission as an intermediate step that investigates general control problems for overactuated systems, while we consider the inclusion of cost terms such as metabolic energy as the focus of future work.
>
> > Is the value of Delta t kept fixed for the bulk of the results?
>
> Delta_t is set to the values described in **Table 4 (Supp.)** for the relevant environments and is otherwise held fixed for all experiments.
>
> > To what extent can the final structure of the matrix C be interpreted?
>
> We have not investigated the specific structure of the C matrix, as DEP is often only active for short periods of time before the policy takes over control or the episode is terminated. We have, however, provided the action correlation matrix of recorded DEP trajectories in **Fig. 13 (Supp.)**. Here, we can see that DEP induces strong correlations and anti-correlations in agonistic and antagonistic muscle groups. These correlations consistently emerge very quickly during learning. See also the new videos on the website: we can see that whenever DEP is active, muscles are contracted in groups to induce motion.
>
> > how critical it is to have knowledge of the 1:1 correspondence between sensors and actuators.
>
> Very! We reran pure DEP exploration on ostrich-run with an inverse model that still connects actuators and muscle length sensors in pairs, but in a randomly shuffled pattern. Note that the inverse model used in the main text is just an identity matrix. It seems that with the modification, no correlated action noise is emerging. Although the assumption of known 1:1 correspondence is a simple one and known in the majority of technical and biological systems, DEP seems to not work well without it.
> We also want to highlight that all muscle-driven systems we consider already have multi-articular muscles, i.e. muscles that span multiple joints at the same time, which does not seem to hinder DEP exploration.

---

> > ### Author Response · Authors · 2022-11-15
> > **Response to sbak (2/2)**
> >
> >
> > > Perhaps call this a first-order decay term, which presumably acts as a regularizer?
> >
> > We have changed the wording in the text.
> >
> > >What forms of perception are available for the last three ostrich tasks, i.e., foraging, stepdown, slopetrotter?
> >
> > For ostrich-foraging, which only requires neck motion, the current and target locations of the beak are included in the state, in addition to the default modalities which include joint and muscle kinematics as well as muscle activity.
> > For the obstacle tasks, slopetrotter and stepdown, NO information of the upcoming obstacles is included in the state, as it is identical to the state of ostrich-run. The policies learn to stabilize the running gait in a purely “blind” and reactionary fashion.
> >
> > > are these actuation dynamics being modeled or not?
> >
> > Activation dynamics are explicitly modeled with MuJoCo default settings as an action and activity dependent low-pass filter. We have stated this now explicitly in **Supp. A.4**.
> >
> > > DEP-MPO for the human hopping is, subjectively speaking for this reader, the highest quality result. Very nice.
> >
> > We thank you very much!
> >
> > Finally, we want to address this point:
> >
> > > or if these results are predictive in any way of what can be observed in natural systems
> >
> > As you have rightfully mentioned, some of the results do not “look” natural yet. We understand this work as a stepping stone in the quest for control of highly overactuated musculoskeletal systems. However, so far we have not included any incentive that would yield natural motion, besides the overall structure of the musculoskeletal systems. It seems that, for this selection of tasks, pure performance does not require natural motion. We hope that this gap can be closed in future work.
> >
> > We thank the reviewer again for their extensive review and hope that they will find the time to read our answers.

---

> > > ### Comment · Reviewer_sbak · 2022-12-03
> > > **Acknowledgement of author response**
> > >
> > > Thank you for the detailed response. The latest changes have further improved the paper.

---

### Official Review · Reviewer_x7yt · 2022-10-25

**Confidence:** 4
**Correctness:** 4
**Technical Novelty And Significance:** 3
**Empirical Novelty And Significance:** 3
**Recommendation:** 8

**Clarity, Quality, Novelty And Reproducibility:**

I believe this paper is well-written and novel. It would be great to release the code to the community, as the authors promised.

**Details Of Ethics Concerns:**

Not Applicable.

**Strength And Weaknesses:**

Strength:
1) This paper is well-motivated and the writing is clear.
2) The idea of using the differential extrinsic plasticity strategy is interesting and it’s novel to adapt it in the reinforcement learning tasks.
3) The authors promised to release the code, which could be a benefit to the whole community.

Weakness:
1) Is there a specific reason of using MPO as the baseline, instead of some other methods like TD3, SAC, PPO?
2) Can the authors provide some qualitivative visualization about what is the learned exploration strategy?

**Summary Of The Paper:**

This paper proposes an exploration strategy for reinforcement learning. This exploration strategy is based on the differential extrinsic plasticity rule in neuroscience which outputs temporal correlated and state-dependent exploration noise. The exploration strategy can also be parameterized as a neural network and this paper update it by using a gradient-free method. During training, it is used as an intra-episode exploration and the overall method achieves large improvement over baselines on overactuated and musculoskeletal systems in simulation.

**Summary Of The Review:**

In summary, I think this paper is a good paper and worth reading for the community. The exploration can be helpful for varies applications such as RL from image observations or RL on more complex systems.

---

> ### Author Response · Authors · 2022-11-15
> **Response to x7yt**
>
> Thank you for the interesting suggestions! We are happy that the effective combination of DEP and RL is seen as novel.
>
> > The authors promised to release the code, which could be a benefit to the whole community
>
> We will provide a curated repository with installation instructions upon acceptance, while the unpolished code is already publicly available in the uploaded supplementary material.
>
> > Is there a specific reason of using MPO as the baseline, instead of some other methods like TD3, SAC, PPO?
>
> In preliminary experiments, we found that MPO performs better in tasks with large action spaces than other common RL algorithms, which is why we chose it for our investigation. In principle, any off-policy algorithm might be used: our main point is that DEP is able to improve exploration for overactuated systems, if the agent suffers from local optima or cannot find sparse rewards.
> In **Fig. 14 middle graph (Suppl.)**, it can be seen that TD4, a distributional variant of TD3, also benefits from DEP, even though the overall performance is better for DEP-MPO.
>
> > Can the authors provide some qualitivative visualization about what is the learned exploration strategy?
>
> That is a nice idea that we hadn’t considered yet! We have added videos for arm26 and humanreacher that show the interplay between the RL policy and DEP at different stages during training and uploaded them to the website. The humanreacher video especially showcases how the DEP response is strongly dependent on the RL behavior that was executed beforehand.
>
> We also want to highlight existing visualizations in the submission version: exploration trajectories in **Fig. 2 (main paper)**, **Fig. 11** and **Fig. 12 (Supp.)** as well as the videos on the **website** which show pure DEP exploration for most of the investigated tasks. We also provided state space visualizations for the continuous action mountain-car environment in **Fig. 19 (Supp.)**.

---

> > ### Comment · Reviewer_x7yt · 2022-11-21
> > **Response**
> >
> > I thank the authors for their detailed response to my review. I’m happy to keep my original recommendation.

---

### Official Review · Reviewer_PfGN · 2022-10-28

**Confidence:** 2
**Correctness:** 4
**Technical Novelty And Significance:** 3
**Empirical Novelty And Significance:** 4
**Recommendation:** 8

**Clarity, Quality, Novelty And Reproducibility:**

This paper's studied issue, idea, and experiments are clearly presented. The studied problem is interesting, and the proposed method is novel. In addition, the implementation details are all in the supplementary. Therefore, I have no problem with reproducibility.

**Strength And Weaknesses:**

+) The idea to integrate differential extrinsic plasticity (DSP), a method from self-organization, into the traditional RL framework is interesting and self-motivated. Although the way of integration (alternating exploration policy between DSP and learned policy) is simple, it tackles the core problem causing ineffective policy learning with overactuated action spaces. Moreover, the studied problem in this work is also essential and appealing.

+) The paper is well-written and easy to follow. It clearly demonstrates its task, challenges (theoretically and empirically), and the proposed idea (with necessary background reviews). The figures clearly illustrate the results qualitatively and quantitatively. The supplementary videos also show the effectiveness of the learned policy with many qualitative examples.

+) The proposed method works well across different agents (torquearm, arm26, ostrich, human arm, human walk) on different tasks (reaching, foraging, walking, hopping). The robustness experiments further show that the learned policy is able to maintain its performance better than baselines.

o) As mentioned by the authors in the conclusion section, the integration of DEP and RL in this work is firmly straightforward, and it might not directly provide much impact on other applications. I am looking forward to seeing more general or sophisticated frameworks, including ideas inspired by neuroscience or biology.

-) I am curious about the sensitivity of the policy to the switch probability p_switch in the Intra-episode exploration stage. In the supplementary (table 4), the p_switch seems hand-picked for different agents for different tasks. Do authors conduct an ablation study regarding it?

**Summary Of The Paper:**

This work studies a novel and interesting problem: muscular control with overactuated action spaces. It first demonstrates that the main issue with overactuated action spaces is how to effectively and efficiently perform exploration using a clear torquearm example with 2 and 600 DoF theoretically and empirically. It further proposes a framework that integrates differential extrinsic plasticity, a method from the domain of self-organization, into the exploration stage of reinforcement learning. The experimental results across torquearm, arm26, human reacher, ostrich-foraging, ostrich-run, human-run, and human-hop show that the proposed framework can learn a better policy with overactuated action spaces compared to various baselines. Moreover, this work also shows that the learned policy is robust to some forms of environmental changes.

**Summary Of The Review:**

This work proposes integrating DSP into the traditional RL framework to tackle the inefficient exploration issue with overactuated action spaces. The idea of utilizing DSP for a more effective exploration is self-motivated. In addition, the studied problem, proposed idea, and the final results in the paper are well-presented. The quantitative results are impressive, and the qualitative results show that the proposed framework can successfully learn a robust policy with overactuated action spaces.

---

> ### Author Response · Authors · 2022-11-15
> **Response to PfGN**
>
> Thank you for this thorough review. We are glad that the extensive application examples, as well as the simplicity of the integration of DEP and RL, are quantitatively and qualitatively convincing.
>
> > I am looking forward to seeing more general or sophisticated frameworks, including ideas inspired by neuroscience or biology.
>
> We agree that this is an exciting and necessary next step to conduct. We hope that even more effective integration methods can be found.
>
> > In the supplementary (table 4), the p_switch seems hand-picked for different agents for different tasks. Do authors conduct an ablation study regarding it?
>
> This is an interesting observation. In general, we find the method to be somewhat robust to the exact value of $p_{\text{switch}}$, as long as it is quite small (<0.1).
>
> Nevertheless, we have conducted an ablation study on humanreacher and ostrich-run and added it in **Fig. 20 (Supp.)**. For humanreacher, the method is not very sensitive to the exact value of $p_{\text{switch}}$. Ostrich-run, the most difficult task, seems to require a more precise adjustment to achieve maximum performance. We conjecture that locomotion is in general much more unstable than reaching and ill-chosen exploration settings can prevent convergence of the RL agent. Each value in the plots is averaged over 5 random seeds; ostrich-run experiments have been trained for $10^{8}$ steps, while humanreacher has been trained for $3\times 10^{7}$ steps.

---

### Author Response · Authors · 2022-11-15
**General Response**

We thank all reviewers for the absolutely amazing recommendations and overwhelmingly positive reviews!
Our work was considered impressive and well-presented (reviewer PfGN), well-motivated and novel (reviewer x7yt), impressive on an important and difficult problem (reviewer sbak) and making a big step forward in solving muscle-driven control problems (reviewer  Bv9J).

After carefully reading all the reviews, we have made the following new contributions and changes:

* We ran ablation studies for the value of $p_{\mathrm{switch}}$ on humanreacher and ostrich-run, which we have added to **Fig. 20 (Supp.)** (reviewer PfGN)
* We provided visualizations of the interplay of DEP and the RL policy on arm26 and humanreacher, shown on the **website**. (reviewer x7yt)
* We ran additional experiments where the inverse model was replaced by a randomly shuffled identity matrix, incorrectly connecting sensors to actuators (**website**), we have made changes to the DEP text (**Sec. 4.1**) and added additional information about the muscle modeling and the activation dynamics (**Supp. A.4**) (reviewer sbak)

All changes are available in the revised manuscript and marked in red.

We again want to thank all the reviewers and the area chair for their valuable time and hope that our responses are satisfying. We will be available for further discussions throughout the rebuttal period.

---

### Decision · Program_Chairs · 2023-01-20

**Decision:**

Accept: notable-top-25%

**Justification For Why Not Higher Score:**

The paper's method is only demonstrated for simulated systems, all on one simulator. Application on "real" high-dimensional actuator systems, beyond the described, would create a fascinating full story.

**Justification For Why Not Lower Score:**

As described in the meta-review.

**Metareview: Summary, Strengths And Weaknesses:**

The paper uses differential extrinsic plasticity to control overactuated agents with RL, and show very efficient learning.  Reviewers agree on the excellence of the paper.


**Note From Pc:**

if the above contains the word "oral" or "spotlight" please see: "oral" presentation means -> notable-top-5% and "spotlight" means -> notable-top-25%. As stated in our emails, we are disassociating presentation type from AC recommendations